



# Reanalysis representation of low-level winds in the Antarctic near-coastal region

Thomas Caton Harrison[1], Stavroula Biri[2], Thomas J. Bracegirdle[1], John C. King[1], Elizabeth C. Kent[2], Étienne Vignon[3], and John Turner[1]

[1]British Antarctic Survey, Cambridge, UK
[2]National Oceanography Centre, Southampton, UK
[3]Laboratoire de Météorologie Dynamique/IPSL/Sorbonne Universités/CNRS, UMR 8539, Paris, France

**Correspondence:** Thomas Caton Harrison (thoton@bas.ac.uk)

**Abstract.** Low-level easterly winds encircling Antarctica help drive coastal currents which modify transport of circumpolar deep water to ice shelves, as well as the formation and distribution of sea ice. Reanalysis datasets are especially important at high southern latitudes where observations are few. Here, we investigate the representation of the mean state and short-term variability of coastal easterlies in three recent reanalyses, ERA5, MERRA-2 and JRA-55. Reanalysed winds are compared

with summertime marine surface wind observations from the ASCAT scatterometer and surface and upper air measurements from coastal stations. Reanalysis coastal easterlies correlate highly with ASCAT (r=0.91, 0.89 and 0.85 for ERA5, MERRA-2 and JRA-55 respectively) but notable wind speed biases are found close to the coastal margins, especially near complex orography and at high wind speeds. To characterise short-term variability, 12-hourly reanalysis and coastal station winds are composited using self-organising maps (SOMs), which cluster timesteps under similar synoptic and mesoscale influences.

Reanalysis performance is sensitive to the flow configuration at stations near steep coastal slopes, where they fail to capture the magnitude of surface wind speed variability when synoptic forcing is weak and conditions favour katabatic forcing. ERA5 exhibits the best overall performance, has more realistic orography and a more realistic jet structure and temperature profile. These results demonstrate the regime behaviour of Antarctica's coastal winds and indicate important features of the coastal winds which are not well characterised by reanalysis datasets.

**1   Introduction**

Easterly surface winds prevail over most of coastal Antarctica and extend up to about 2 km above surface level. Easterly wind stress is in turn an important driver of the westward Antarctic Coastal Current, a circulation coupled in many regions to the Antarctic Slope Front (ASF) between shelf waters and warmer circumpolar deep water (Thompson et al., 2018). The Southern Ocean meridional overturning circulation may respond more to changes in Antarctic coastal winds (or 'polar easterlies') than

to the westerlies further north (Stewart and Thompson, 2012). It has been shown that modifications to easterly wind stress and Ekman pumping near the coast could enhance subsurface advective heat fluxes across the ASF, contributing to ice shelf basal melt (Spence et al., 2014; Goddard et al., 2017; Stewart et al., 2019), while coastal katabatic winds could act as a control on ice shelf stability in some regions (Hazel and Stewart, 2020). Coastal winds also modify sea ice concentrations, for example





off the Ross Ice Shelf (Kurtz and Bromwich, 1985; Petrelli et al., 2008; Mathiot et al., 2012; Silvano et al., 2020). Given the
sparsity of nearby observations, reanalysis datasets are critical for characterising Antarctic coastal winds, and often provide
boundary conditions for atmospheric and ocean models. As an observational benchmark they are also important for evaluation
of coupled models used for future projections.

In part due to the paucity of assimilated observations, historical trends of the coastal easterlies from reanalysis are uncertain
and differ among datasets, though all indicate an increase in their seasonality over the last few decades (Hazel and Stewart,
2019). Over Antarctica itself, the magnitude of the late $20^{th}$ century trend in surface winds differs considerably in its pattern
and magnitude between reanalyses, especially in the coastal region (Dong et al., 2020). On average, reanalyses exhibit higher
wind speed correlation coefficients with surface station measurements in summer compared to winter and over inland regions
compared to the coast (Tetzner et al., 2019; Dong et al., 2020). The realism of tropospheric wind profiles is improved in the
more recent ERA5 reanalysis with respect to the earlier ERA-Interim at coastal stations, but large deficiencies remain (Vignon
et al., 2019). Reanalysis evaluations have tended to focus on the onshore coastal region, in part due to the limited availability
of satellite observations near the coast (Bourassa et al., 2019), but, since offshore winds play a key role in interactions between
the atmosphere, ocean and cryosphere, an evaluation of performance against observations from the marine coastal sector is
needed.

Alongside the lack of evaluation offshore, relatively little attention has been given to reanalysis representation of wind
variability and extremes. Jones et al. (2016) show how considerable underestimation of low-level wind speeds occurs in the
Amundsen Sea Embayment when a 'low-level jet' is present, reminiscent of underestimated elevated wind speeds in regions
of complex orography in the Arctic (e.g., Moore et al., 2016). Extreme coastal winds are important for rapid sea ice loss (Jena
et al., 2022) and calving events (Francis et al., 2020) so a characterisation of reanalysis performance during high-wind states is
needed. However, Antarctic coastal winds are influenced by a range of drivers, each of which presents a different set of chal-
lenges for reanalysis representation. Although mean coastal wind speeds peak onshore, the variability on 12-hourly timescales
is highest just offshore where the interaction between directionally-constant continental winds and highly variable synop-
tic flows becomes important (Figure 1). One important driver is katabatic forcing, which sustains shallow terrain-following
drainage flow towards the coast (Ball, 1960; Parish and Bromwich, 1987) but its offshore extent is a source of uncertainty
as the flow often stops abruptly at the coastal margin (e.g., Yu and Cai, 2006; Tomikawa et al., 2015; Vignon et al., 2020)
and its behaviour in models is sensitive to the representation of the atmospheric boundary layer (King et al., 2001; Parish and
Cassano, 2003; Orr et al., 2014). Representation of orography is important as long-wave cooling over the steep coastal slopes
sets up both katabatic flows near the surface and a deeper layer of terrain-following isentropes above it which explains the
easterly 'low-level jet' structure (Fulton et al., 2017). Orography is also critical for blocking and barrier winds induced by
steep Antarctic coastal slopes (Parish and Cassano, 2001, 2003; Orr et al., 2014; Weber et al., 2016; Yamada and Hirasawa,
2018). Other drivers include the large-scale pressure gradient force (van den Broeke and van Lipzig, 2003) and, on shorter
timescales, synoptic storms.

In this paper, we aim to provide both a representative overview of reanalysis performance in the coastal easterly sector but
also to test whether that performance is sensitive to short-term variations in the large-scale pressure conditions driving the



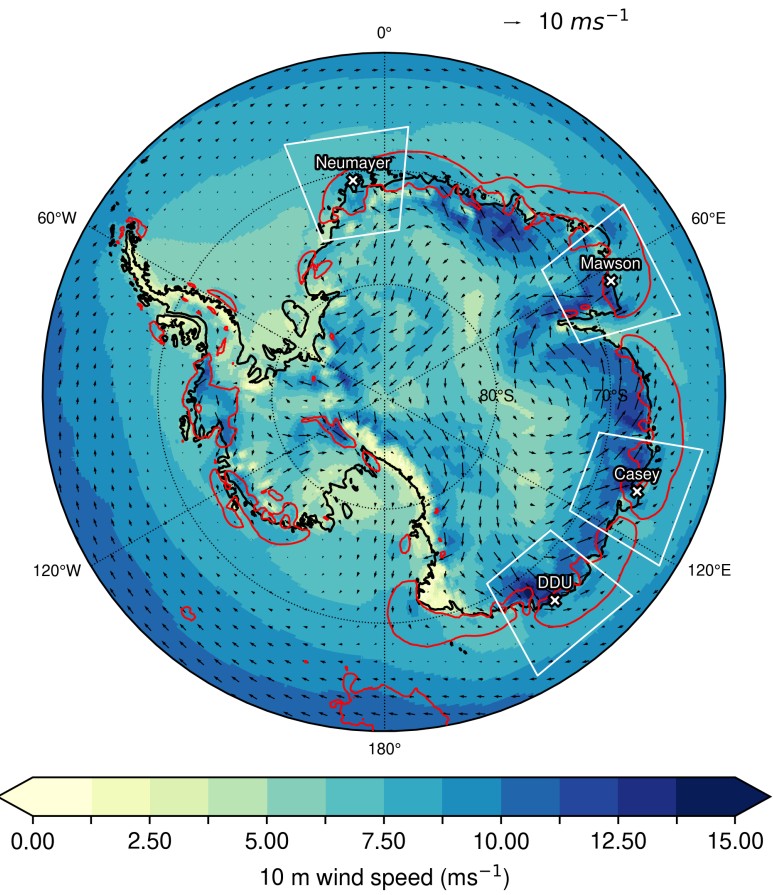

**Figure 1.** Mean ERA5 10 m wind speed for the period January 2010 to December 2017 (shaded). The 4 ms$^{-1}$ contour of the standard deviation of 12-hourly 10 m wind speed for the same period is overlain in red. White boxes indicate the regions selected for the SOM analysis, with labelled station locations.

coastal winds. By comparing reanalysis performance across driving states and geographic locations, we aim to shed light on possible sources of error and offer a process-oriented perspective to help select the most appropriate reanalysis to use when characterising the Antarctic coastal easterlies. The research in this paper is guided by three questions:

1. How well do reanalyses represent the spatial patterns and short-term variability of surface winds within the Antarctic coastal easterly sector?

2. To what extent is reanalysis representation of Antarctic coastal winds sensitive to variations in the driving flow regime?



3. Which reanalysis most accurately represents the Antarctic coastal easterlies?

Our analysis has two components; firstly, to quantify reanalysis performance across the entire coastal region we compare with observations from the EUMETSAT MetOP-ASCAT sensor, which has near-coastal data available during austral summer. Next, to evaluate reanalysis representation of variability on short timescales, a self-organising map (SOM) methodology is used to composite surface and upper air wind data available from four coastal surface stations which have high-resolution

radiosonde data compiled by Vignon et al. (2019).

Section 2 details the datasets used as well as the collocation and SOM approach. Results are presented in Section 3, including an evaluation against ASCAT (3.1) and a SOM-based evaluation against station measurements (3.2). Results are discussed and an overall assessment of reanalysis reliability is given in Section 4 with conclusions in Section 5.

## 2    Data and Methods

### 2.1    Reanalysis datasets

For this analysis, we evaluate ERA5, MERRA-2 and JRA-55, which belong to the latest generation of reanalyses which have been widely adopted for studies of the Antarctic circulation as well as for quantifying historical trends and evaluating climate models.

### 2.1.1    ERA5

ERA5 is the fifth generation in the European Centre for Medium-range Weather Forecasts (ECMWF) series of reanalyses. A full technical description of the setup is given by Hersbach et al. (2020). The dataset is derived using a 2016 version of the operational ECMWF Integrated Forecasting System (IFS) model (cycle 41r2) and a hybrid incremental 4D-Var data assimilation methodology. ERA5 is available on a regular $0.25°$ grid at 31 km horizontal grid spacing and on 137 vertical levels. Both surface-level diagnostics and data on pressure levels are available hourly. A final release of ERA5 is publicly

available for 1959 onwards. ERA5 surface winds are a diagnostic derived in IFS output by scaling the winds at the 40 m height level to 10 m assuming a roughness length for short grass (0.03 m) when the surface roughness is less than or equal to 0.03 m and otherwise using the tile roughness length. For our comparison with ASCAT ocean winds, the ERA5 10 m neutral wind product is used instead, which is derived from surface stress using the tile roughness length and assuming neutral stability.

### 2.1.2    MERRA-2

The Modern-Era Retrospective Analysis for Research and Applications, version 2 (MERRA-2) has been developed by NASA's Global Modeling and Assimilation Office (GMAO) as an intermediate reanalysis intended to incorporate updates to GMAO modelling and data assimilation into the MERRA framework (Gelaro et al., 2017). MERRA-2 is based on the Goddard Earth Observing System-5 (GEOS-5) forecast model and is available from 1979 on an approximately $0.5 \times 0.625°$ grid (50 km grid spacing) with 72 vertical levels. Surface fields (including 10 m wind speed) are available hourly, with 3-hourly vertical fields



on pressure levels. Wind speeds at 10 m are derived in MERRA-2 by interpolating from the lowest model level using Monin-Obukhov similarity theory, thereby accounting for the effects of varying near-surface atmospheric stability (Torralba et al., 2017). These stability-dependent winds are compared with station measurements. For comparison with ASCAT, we calculated 10 m neutral wind speeds using the hourly output of friction velocity ($u_*$) and roughness length ($z_{0m}$) from the MERRA2 flux diagnostics (GMAO, 2015) following:

$$\mathrm{u}_{10\mathrm{n}} = \frac{u_*}{k} \ln\left(\frac{10}{\mathrm{z}_{0\mathrm{m}}}\right) \tag{1}$$

where $k$=0.4 is the von Kármán constant.

### 2.1.3 JRA-55

The Japanese 55-year Reanalysis (JRA-55) developed by the Japan Meteorological Agency (JMA) covers a period from 1958 onwards and is an update to JRA-25, incorporating novel observations and a 4D-Var data assimilation system (Kobayashi
et al., 2015). The JMA global spectral model (GSM) is used as the underlying forecast model. The JRA-55 grid corresponds to approximately 55 km horizontal grid spacing, with 60 vertical levels. Surface fields are available on six-hourly timesteps, so compared to ERA5 and MERRA-2 the comparison with relatively infrequent scatterometer observations is more limited (Section 2.2.1). JRA-55 10 m winds are interpolated from the lowest model level as in MERRA-2, but unlike the stability-dependent ERA5 and MERRA-2, surface winds are calculated assuming neutral stability in the surface layer (Torralba et al.,
110 2017).

### 2.2 ASCAT

Remote Sensing Systems (RSS, http://www.remss.com) use radar backscatter data from the EUMETSAT MetOP-ASCAT sensor to derive surface wind vectors at 10 m height above sea surface over ice-free open water surfaces from 01/03/2007 onwards on a $0.25° \times 0.25°$ Cartesian grid (Ricciardulli and Wentz, 2016) with good coverage of polar latitudes. The open-ocean
backscatter data are converted into winds using a Geophysical Model Function (GMF). Advanced Scatterometer (ASCAT, version 2.1) is a C-band scatterometer used to measure surface radar backscatter along two swaths each ∼500 km wide separated by a 360 km gap processed to provide wind vectors for a grid spacing of about 25 km. The data contain a rain flag, since rain can produce a positive bias at low wind speeds due to backscatter from rain drops and negative bias at high wind speeds due to atmospheric attenuation of the signal. Undetected sea ice can limit the quality of scatterometer data as well as wind speeds
higher than 35 ms$^{-1}$ (Ricciardulli and Wentz, 2016). Rivas and Stoffelen (2019) report that the magnitude of ASCAT wind component errors on the scatterometer measurement globally is ∼0.7 ms$^{-1}$ with negligible bias. ASCAT wind observations are assimilated into all three reanalyses used in this paper from 2008.

### 2.2.1 Collocation of reanalyses and ASCAT

For comparison, the reanalysis winds were mapped to the ASCAT grid by selecting the nearest neighbour in space and time
and requiring a time match within an hour. To collocate spatially, we remap the reanalysis data onto the ASCAT grid using





a nearest neighbour interpolation scheme. Every ASCAT swath pair has a different overpass time with the observation times of ascending and descending pass segments being interleaved throughout the day. For the collocation in time, we first round the time of each ASCAT observation to the nearest full hour and store these values as an array. Then for each grid point we collocate the rounded time step in the array with the timestep in the reanalysis for the same point. The time resolution for

JRA-55 is coarser (6 hourly), thus the rounded ASCAT timestep can only be "matched" on four timesteps (i.e. 00, 06, 12, 18 hours), and as a result certain overpasses cannot be collocated following our time-collocation methodology. The points that cannot be collocated are left missing.

Analysis is primarily constrained to austral summer due to the seasonal expansion of sea ice which prevents wind speed observations near the coast. A breakdown of the results by season is given in Appendix D.

## 2.3    In-situ wind observations

Surface (10 m) and upper air (radiosonde) wind observations from four representative permanent East Antarctic sites are used. These include Neumayer on the Ekström Ice Shelf and three stations at the foot of steep coastal slopes, Mawson, Casey and Dumont d'Urville (DDU) (Figure 1). 10 m wind speed measurements from surface stations are withheld from assimilation into JRA-55, MERRA-2 and ERA5 (Kobayashi et al., 2015; Gelaro et al., 2017; Hersbach et al., 2020), but the local wind field is

not entirely independent from observations due to the assimilation of upper air wind, pressure and temperature profiles. Surface meteorological observations were obtained via the SCAR READER database (Turner et al., 2004). Neumayer, Mawson and DDU were selected for this analysis to evenly sample the East Antarctic coastline as well as due to their high representativeness of the coastal lower troposphere (see Vignon et al. (2019), Figure 11). Casey, on the other hand, is less representative but is situated near complex orography which is represented differently in the three reanalyses (see Section 2.3.1). Long-term vertical

profile data from West Antarctica are currently lacking but would be highly valuable for an evaluation of reanalysis performance of this kind, for example under various configurations of the Amundsen Sea Low.

High resolution vertical wind profiles measured with Vaisala RS-92 radiosondes (Modem M2K2-DC at DDU) from the dataset collated by Vignon et al. (2019) are used, with technical details described in Section 2.1 therein. Important post-processing steps include calculation of measurement heights using a standard hydrostatic approximation based on temperature,

humidity and pressure readings (König-Langlo et al., 1998), followed by linear interpolation to 10 m resolution from data available at 1 second intervals at Mawson, DDU and Casey and 5 second intervals at Neumayer. Following Vignon et al. (2019), the lowest 100 m of data are not used due to possible thermal lag error (e.g. the radiosonde not equilibrating with outdoor temperatures prior to launch) and to allow the balloon to reach ambient flow velocity.

### 2.3.1    Collocation of reanalyses and station measurements

Comparison of gridded datasets with point observations is not straightforward, especially in regions of complex orography or where steep slopes generate an offset between the actual station elevation and that at the station location in the model (Dong et al., 2020). In this evaluation we compare station observations with the nearest reanalysis grid point for both the surface and upper air data as in a number of previous analyses (Jones et al., 2016; Gossart et al., 2019; Tetzner et al., 2019; Vignon et al.,





**Figure 2.** Three left columns (ERA5 leftmost, MERRA-2 in the centre and JRA-55 the rightmost) show correlations between 10 m wind speed measured at surface stations with the local 10 m reanalysis wind field, with the selected nearest neighbour reanalysis gridpoint outlined in white. Rightmost column shows south-north cross sections through the location of the station, with orography from the RAMP2 dataset shaded in grey and reanalysis orography from ERA5 (red), MERRA-2 (blue) and JRA-55 (green) indicated as a line. Filled markers on the lines mark the respective nearest neighbour reanalysis gridpoint. Shown are Casey (a-d), DDU (e-h), Mawson (i-l) and Neumayer (m-p).



2019). Within the lowest 3000 m, sondes do not generally drift more than a few kilometres horizontally (Vignon et al., 2019,
Appendix C) so we do not account for sonde drift in this analysis. We recognise that the chosen gridpoint may not be fully
representative of the local conditions at the station, however. To explore this further, we map the correlation between measured
10 m surface winds and all nearby grid points (Figure 2).

Correlations between reanalysis and station 10 m wind speeds (Figure 2, three left columns) are high over a large region
surrounding most stations, suggesting local factors do not dominate the observed winds so a comparison with coarser gridded
data is meaningful. A possible exception to this is Casey (Figure 2a-c), where although correlations close to the station are
high, shifting the selected reanalysis grid point slightly could have a large impact upon results. This is likely due to the Law
Dome east of Casey (66°S, 112° E) which, at a peak altitude of 1,395 m, is both a barrier to the large-scale flow and a likely
source of topographically induced gravity waves with highly localised effects (Murphy and Simmonds, 1993; Turner et al.,
2001; Adams, 2004). All reanalyses include some representation of the Law Dome but in ERA5 the peak rises to 1,259 m
compared to 964 m in MERRA-2 and 952 m in JRA-55.

Orographic slopes from the plateau to the coast differ considerably between reanalyses (Figure 2, rightmost column). All
three datasets have orography that is noticeably smoothed compared to the 1 km Radarsat Antarctic Mapping Project Digital
Elevation Model (DEM) version 2 (RAMP2) (Liu et al., 2015). ERA5 orography is derived from RAMP2 south of 60°S and
so is closest to this benchmark but in general the slope gradient is not as steep meaning the altitude of the matched grid point
is greater than the station altitude along the coastal slopes and approximately equal (within 60 m) at Neumayer on the flat
Ekström Ice Shelf.

To test the sensitivity of the results to grid point collocation, the analysis was repeated with station wind data matched to each
of the two grid points immediately north and south of the nearest neighbour (i.e. the grid box outlined in white in the three left
columns of Figure 2), with results described in Appendix A. In brief, the value of the bias for states with a primarily katabatic
influence is very sensitive to this collocation test due to the sharp cutoff which occurs at the coast, but other performance
characteristics and the differences between stations are robust.

## 2.4  Self-organising maps

Wind field variability local to observing stations is characterised in this study using self-organising maps (SOMs). The goal is
to group periods with similar local pressure conditions, including the magnitude, gradient and orientation, without needing to
create a-priori metrics to distinguish them. SOMs are a data-driven approach used to cluster multidimensional data into similar
groups arranged in a 2D grid with a user-specified number of 'nodes'. A summary of the unsupervised learning algorithm
applied in a climate context is given by Skific and Francis (2012). The iterative SOM approach allows non-linearities in the data
distribution to be accounted for, compared for instance to a principal component analysis which produces linear combinations
of features in the data space (when the important underlying patterns may not in fact vary linearly). The main advantage of
SOMs over other similar techniques is that nearby nodes are updated during training such that in the final set of nodes similar
patterns or states are grouped together. Examples of use in an Antarctic context include the study of patterns and drivers of
the Ross Ice Shelf Airstream (Nigro and Cassano, 2014b, a) and evaluation of the Antarctic Mesoscale Prediction System





(AMPS) against station data (Jolly et al., 2016), with similar clustering techniques used to classify synoptic states associated with Antarctic surface melt episodes (Scott et al., 2019).

SOMs in our analysis are driven using the 12-hourly ERA5 mean sea level pressure (MSLP) field over a $200 \times 200$ km grid centred upon the location of each station for 2010-2017. Each 12-hourly MSLP field in ERA5 is then matched to the closest SOM node (lowest squared difference) to determine a set of dates for each SOM. Observations and reanalysis winds are then composited by these dates. 12-hourly data are used to match the approximate frequency of sonde launches at coastal stations. SOMs are calculated using the Somoclu library for Python: https://somoclu.readthedocs.io/en/stable/download.html (Wittek

et al., 2017).

Several metrics are used to determine the appropriate number of SOM nodes. With fewer nodes, the SOM representation of synoptic states may be too generalised. Conversely, with a larger array duplicates are introduced (Cassano et al., 2015; Gibson et al., 2017) though how undesirable this is depends on the application (Hewitson and Crane, 2002). The similarity among SOM nodes, similarity of the composited wind fields and correlations between data points (in this case 12-hourly MSLP fields)

and their respective SOMs are quantified for various configurations in Figure B1. Appendix B also provides further details on the selection of SOM configuration. Here, we use 6 SOM nodes (organised onto a $2 \times 3$ grid) which represents approximately four states with relatively strong synoptic forcing and two with weaker forcing and conditions favouring katabatic low-level winds of continental origin.

The period 2010-2017 is used in this analysis for several reasons. Firstly, Neumayer station is in a fixed position on the ice

shelf throughout this period (Neumayer Station III data available since February 2009). Secondly, surface anemometer instrumental models are mostly consistent through this period (Synchrotac 706 series at Mawson and Casey, Thies at Neumayer). Thirdly, ASCAT scatterometer observations are assimilated into the three analysed reanalyses from 2008 onwards (Koster et al., 2016; Hersbach et al., 2020; Kobayashi et al., 2015). Lastly, high resolution upper air winds measured with Vaisala RS-92 radiosondes (Modem M2K2-DC at DDU) are available through this period as used in Vignon et al. (2019).

## 3 Results

### 3.1 Performance against ASCAT

Here we compare the ASCAT 10 m wind dataset with collocated reanalysis 10 m neutral winds from the 2010-2017 sampling period. A breakdown of the results by month is given in Appendix D, which shows how the availability of ASCAT data from the near-coastal region is affected by the presence of sea ice. The fullest coverage comes from the late-summer January to

March period when most of this sea ice has melted. Statistics and scatterplots shown in Figure 3 are calculated from the region of the coastal easterlies demarcated in red in Figure 4, defined as where either the mean ASCAT zonal wind for the period 2010-2017 is less than zero (i.e. easterly) or the gridpoint is within a 12-gridbox buffer zone drawn around the ERA5 land-sea mask coastline. Henceforth this is referred to as the coastal easterly domain.

Values of the ASCAT-reanalysis Pearson correlation coefficient within the coastal easterly domain are highest in ERA5 at

0.91, but MERRA-2 and JRA-55 surface winds are also closely correlated with ASCAT at 0.89 and 0.85, respectively. ERA5



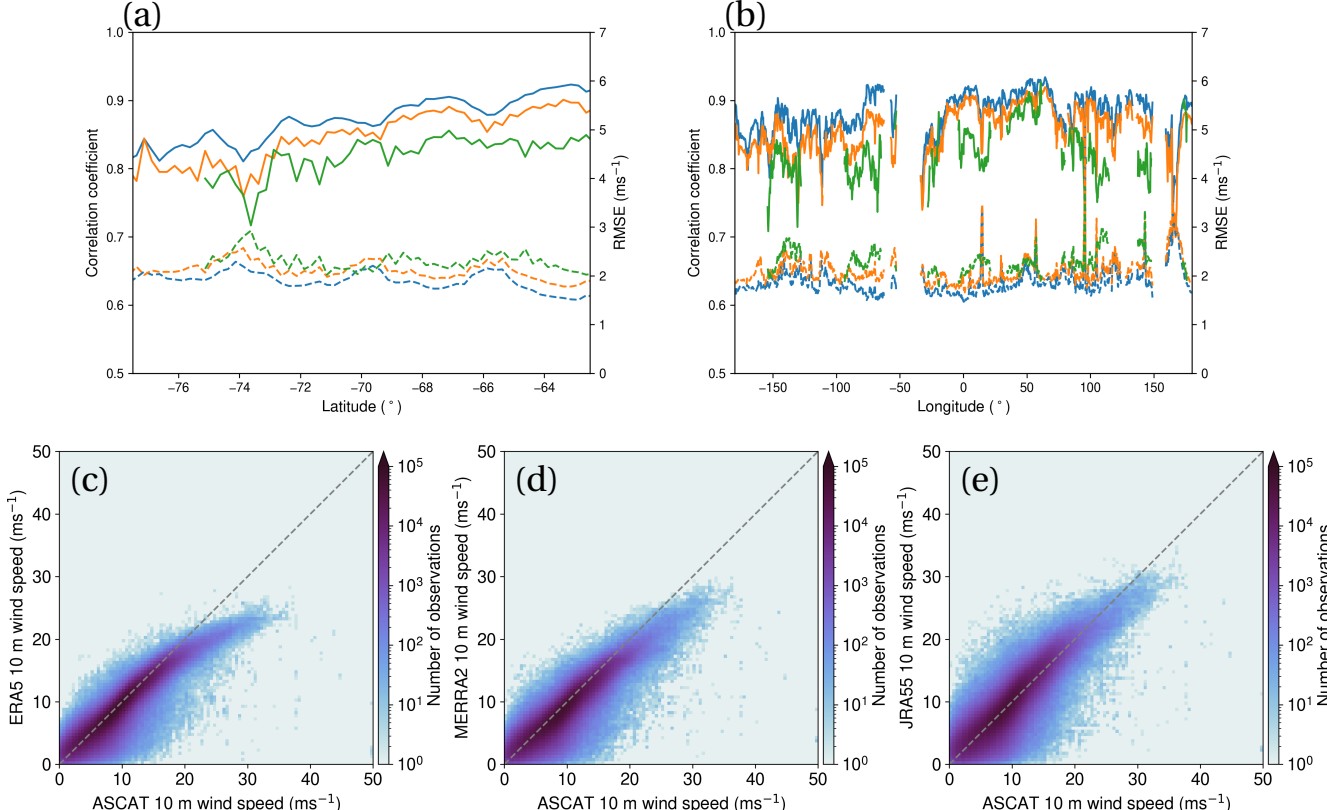

**Figure 3.** (a, b) Mean correlation coefficient (solid lines) and RMSE (dashed) against collocated ASCAT 10 m wind speeds calculated for all data points within the coastal easterly domain for ERA5 (blue), MERRA-2 (orange) and JRA-55 (green). (a) is averaged across longitude bands whereas (b) is averaged across latitude bands. (c)-(e) are heat map scatterplots of collocated data points within the coastal easterly domain for (c) ERA5, (d), MERRA-2 and (e) JRA-55. Only data points collocated in all three reanalyses are included in (c)-(e). Data points which are rain flagged or have a GMF-matchup flag value over 2 are not included.

correlation coefficients are consistently highest around the coast (Figure 3b) but longitudinal variations in performance are very similar between reanalyses. Correlation coefficients decline and RMSE increases with increasing proximity to the coast itself (i.e. higher southern latitudes) (Figure 3a). Scatterplots (Figures 3c-e) indicate a high density of points clustered close to the 1:1 line (dashed grey) between 0 and 20 ms$^{-1}$ though JRA-55 exhibits a higher spread of overestimated wind speeds. All three

reanalyses exhibit a tendency towards negative bias with respect to ASCAT at high wind speeds (above 20 ms$^{-1}$). This may in part relate to the localised and short-term nature of extreme winds observed with the scatterometer. It is also possible that the representation of extremes is affected by the assimilation method; for example the assimilation of extreme scatterometer winds into ECMWF analyses is sensitive to the data thinning and quality control procedures (De Chiara et al., 2017). Underestimation of high wind speeds is also a characteristic of ERA-Interim when compared with in-situ wave glider wind observations from the



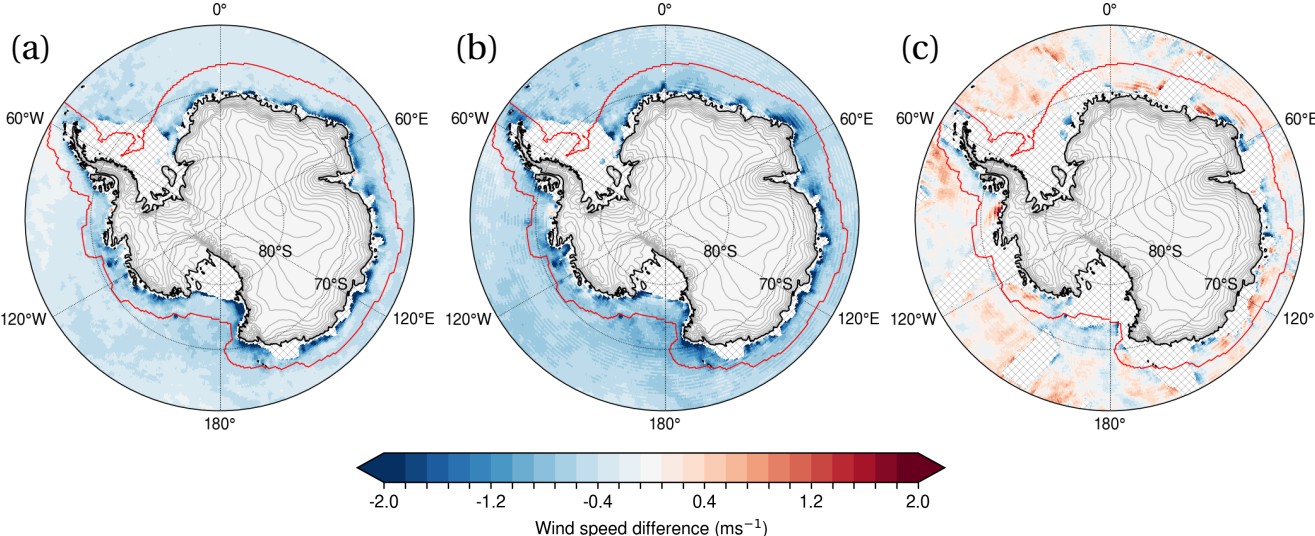

**Figure 4.** Mean reanalysis minus ASCAT 10 m wind speed for the period 2010-2017 for (a) ERA5, (b) MERRA-2 and (c) JRA-55. Hourly data are used for ERA5 and MERRA-2 with six-hourly from JRA-55. Data points which are rain flagged or have a GMF-matchup flag value over 2 are not included and pixels with fewer than 50 ASCAT-reanalysis collocations are masked (hatched region). Orography at 300 m intervals (from the ERA5 invariant fields) is marked with grey contours. The region within the red contour is the coastal easterlies domain, identified where either the ERA5 2010-2017 mean zonal wind is less than zero or the location is within a 12-gridbox buffer drawn from the coast.

sub-Antarctic Southern Ocean by Schmidt et al. (2017). The mean bias across the coastal easterly domain for ERA5, MERRA-2 and JRA-55 is -0.51 ms$^{-1}$, -0.72 ms$^{-1}$ and -0.06 ms$^{-1}$ respectively, but sampling only wind speeds above 20 ms$^{-1}$ this bias changes by -3.89 ms$^{-1}$, -3.88 ms$^{-1}$ and -1.54 ms$^{-1}$ (i.e. becomes more negative). For reference, on average across all the 10 m surface station wind speed data from Mawson, Neumayer and Dumont d'Urville (the better exposed sites), this change in the bias induced by sampling only wind speeds above 20 ms$^{-1}$ would be -4.65 ms$^{-1}$, -5.13 ms$^{-1}$ and -4.53 ms$^{-1}$, respectively.

Maps of the reanalysis bias with respect to ASCAT (Figure 4) indicate that the largest differences between the reanalyses and observations are found in the near-coastal region. Both ERA5 and MERRA-2 exhibit a band of negatively biased wind speeds close to the coastal margins. Some regions of elevated bias are found close to complex or steep orography. For example, ERA5 and MERRA-2 exhibit stronger negative biases along Enderby Land at 50-55° E, a site identified by Sampe and Xie (2007) as a likely hotspot of orographically-driven winds. The western Antarctic Peninsula (west of 60° W) is a region of underesti-

mated wind speeds in all three reanalyses, and underestimation of winds north of the Ross Ice Shelf along the Transantarctic Mountains (from about 65-75°S and 170° E) is found in ERA5 and MERRA-2. JRA-55 does not have as consistent a sign of difference in wind speed compared to the other two reanalyses. Near-coastal negative wind speed biases increase in ERA5 and MERRA-2 from January to March (Figures D2 and D3a-c) as wind speeds over the region increase (Figure D1a-c).





It is interesting to note that although the selected domain has an overall easterly low-level wind regime, biases in the zonal
and meridional components of ERA5, MERRA-2 and JRA-55 10 m winds are mixed around the coast (Figure E1). The
offshore easterly sector has much lower mean directional constancy than the onshore sector (van den Broeke and van Lipzig,
2003) meaning surface wind directions are more variable. As a result, biases may cancel out to an extent when individual wind
components are calculated if the reanalysis bias is insensitive to wind direction.

Our analysis agrees with that of Carvalho (2019) who shows increasing reanalysis errors with respect to satellite wind
products near the poles. Carvalho (2019) also finds that MERRA-2 generally outperforms JRA-55, CFSR, JRA-55 and ERA-
Interim winds at high southern latitudes. We find that the more recent ERA5 now slightly outperforms MERRA-2 in the coastal
region, consistent with the findings of Rivas and Stoffelen (2019) who report an improvement in ERA5 ocean winds relative to
ERA-Interim.

Some of the observed differences between ASCAT and the reanalyses may be due to inconsistencies in how scatterometer
and reanalysis 10m winds are derived, although the effects of atmospheric stability are accounted for in our analysis. One such
issue is mismatches in the designation of open ocean and sea ice in the reanalysis and scatterometer datasets; some regions of
non-zero ERA5 sea ice concentrations were found within the collocated ASCAT fields. Excessively smooth sea ice distributions
in the marginal ice zone have been linked to increased wind speed RMSE with respect to aircraft observations at high northern
latitudes (Renfrew et al., 2021). To estimate the effect of mismatched sea ice, the evaluation of ERA5 was recalculated with
the regions of non-zero ERA5 sea ice concentration masked out. With this masking applied, the overall mean bias in coastal
easterly domain was reduced from -0.67 to -0.59 ms$^{-1}$ with some error-prone regions around the edge of the ice pack removed
(not shown), but the correlation coefficient only increased from 0.91 to 0.92.

Ocean currents may also have an effect on the results as ASCAT measures surface wind stress relative to a moving surface,
but within the region of the easterlies this effect is expected to be small with an estimated mean surface speed south of 65°S
for the 2010-2017 period of 0.007 ms$^{-1}$ from the OSCAR Surface Currents dataset (ESR, 2009). The analysis of Rivas and
Stoffelen (2019) includes a correction for ocean currents which reduces a positive zonal wind bias north of the coastal easterly
domain in the Southern Ocean (i.e. too strong westerlies) in ERA5, with the magnitude of the effect around 0.1 ms$^{-1}$ (see
Figure 10 therein). The effects of air density differences on the estimation of neutral winds are also expected to have a small
impact on the results. Accounting for this could reduce reanalysis biases by about 0.1 ms$^{-1}$ in the near-coastal region based
upon the analysis of de Kloe et al. (2017) (see Figure 13). In summary, these sources of inconsistency do have a small impact
but are not able to explain large wind speed biases exceeding 2 ms$^{-1}$ such as those found close to complex Antarctic orography.

### 3.2 State-dependent performance against station measurements: SOM regimes

### 3.2.1 Composited synoptic and katabatic conditions

Coastal stations are exposed to a variety of synoptic and mesoscale flow regimes, with the largest short-term variability in
winds observed offshore (Figure 1). SOM nodes for each station organise broadly into states for which the pressure gradient
is more intense and ones where it is reduced (Figure 5 and Table 1). However, the SOM nodes are not a simple continuum



between those two extremes; a number of nodes represent the varied orientation of synoptic pressure gradients relative to the location of the station. For example, at Dumont d'Urville (Figure 5b), nodes (1, 1) and (1, 2) have comparable large-scale pressure gradients but the low-level pressure contours associated with the offshore low are orientated along the coast for (1, 1) whereas in (1, 2) they are oriented more perpendicular to the coast. The strongest 10 m wind speeds near each station are observed during states with the largest synoptic pressure gradient ((1, 2) at Casey, (1, 1) at Mawson and Neumayer and both at DDU). The maximum in wind speed is generally found onshore, which could be an indication of combined katabatic and synoptic forcing (e.g., Turner et al., 2009; Orr et al., 2014) but is also the region of maximum large-scale pressure gradient force on average (van den Broeke and van Lipzig, 2003).



**Table 1.** Description of meteorological conditions associated with each SOM node.

Casey

| (0, 0) | (0, 1) | (0, 2) |
|---|---|---|
| Weak pressure gradient. Moderate southeasterlies onshore and moderate easterlies offshore. | High pressure, reduced cloud fractions, strongly favourable for katabatic forcing. Strong southerlies onshore with weak easterlies offshore. | Moderate pressure gradient, moderately favourable for katabatic forcing. Strong southeasterlies onshore with weak easterlies offshore. |
| (1, 0) | (1, 1) | (1, 2) |
| Weak pressure gradient. Unfavourable for katabatic forcing. Moderate southeasterlies onshore and near-zero easterlies offshore. | Strong pressure gradient directing flow into the Law Dome. Strong winds onshore and moderate winds offshore. | Strong pressure gradient directing flow partially around the Law Dome. Tip jet formation and favourable conditions for katabatic forcing. Strongest winds onshore and offshore. |

Dumont D'Urville

| (0, 0) | (0, 1) | (0, 2) |
|---|---|---|
| Moderate pressure gradient. Moderate southeasterlies onshore and strong easterlies offshore. Favourable katabatic conditions. | High pressure, reduced cloud fractions, strongly favourable for katabatic forcing. Strong southerlies onshore with weak easterlies offshore. | Weak pressure gradient, reduced cloud, favourable for katabatic forcing. Strong southerlies onshore with near-zero easterlies offshore. |
| (1, 0) | (1, 1) | (1, 2) |
| Weak pressure gradient. Moderate southeasterlies onshore and strong easterlies offshore. Favourable katabatic conditions. | Strong pressure gradient directing flow along the coast east of DDU. Strong winds onshore and offshore. | Strong pressure gradient directing flow into the coast. Strong winds onshore and offshore. |

Mawson

| (0, 0) | (0, 1) | (0, 2) |
|---|---|---|
| Moderate pressure gradient. Moderate southeasterlies onshore and moderate easterlies offshore. | High pressure, strongly favourable for katabatic forcing. Strong southerlies onshore with weak easterlies offshore. | Strong pressure gradient directing flow into the coast. Strong winds onshore and offshore. |
| (1, 0) | (1, 1) | (1, 2) |
| Weak pressure gradient. Favourable for katabatic forcing. Moderate southeasterlies onshore and near-zero easterlies offshore. | Strong pressure gradient directing flow along the coast east of Mawson. Strong winds onshore and offshore. | Strong pressure gradient directing flow from the southeast. Strong winds onshore and moderate winds offshore. |

Neumayer

| (0, 0) | (0, 1) | (0, 2) |
|---|---|---|
| Weak pressure gradient directing flow from the south. Weak winds onshore and offshore. | High pressure. Weak winds onshore and near-zero winds offshore. | Moderate pressure gradient directing flow along the coast east of Neumayer. Moderate winds onshore and offshore. |
| (1, 0) | (1, 1) | (1, 2) |
| Strong pressure gradient directing flow from the south. Moderate winds onshore and offshore. | Strong pressure gradient directing flow from the northeast. Strong winds onshore and offshore. | Strong pressure gradient directing flow along the coast east of Neumayer. Moderate winds onshore and offshore. |

Each station has two or three nodes which correspond to reduced pressure gradients (Table 1), including one with high pressure onshore (node (0, 1) in Figure 5). To assess the likely contribution of katabatic flow to these states, the acceleration term due to katabatic forcing $K$ (KAT index, hereafter) is calculated for each ERA5 grid point following the approach described in van den Broeke and van Lipzig (2003). Our implementation of this approach is detailed in Appendix C.



**Figure 5.** Composited ERA5 mean sea-level pressure (MSLP) with 10 m wind vectors for each node at (a) Casey, (b) DDU, (c) Mawson and (d) Neumayer from SOMs calculated for the period 2010-2017. Station locations are marked as a white cross. Nodes with limited synoptic forcing but favourable conditions for katabatic forcing are labeled with a [K].

Composites of the katabatic index (Figure 6) reveal that the exposure to conditions favouring katabatic forcing varies considerably between stations. The highest index values are observed to the east of Dumont d'Urville, where the nearby steep coastal slopes favour considerable local baroclinicity. For all stations, the largest nearby composited KAT index values are for node (0, 1), i.e. during high pressure conditions. These nodes are also associated with the lowest mean total cloud cover in ERA5 (not shown), favouring long-wave cooling and a stable boundary layer, and exhibit a sharp cutoff in wind speeds at the coast. Nodes which exhibit these signs of strong katabatic influence but little synoptic influence (i.e. a low background pressure gradient)





**Figure 6.** Composited ERA5 katabatic acceleration term with 10 m wind vectors overlain for each node at (a) Casey, (b) DDU, (c) Mawson and (d) Neumayer from SOMs calculated for the period 2010-2017. Station locations are marked as a white cross. Nodes with limited synoptic forcing but favourable conditions for katabatic forcing are labeled with a [K].

will be referenced several times in the reanalysis evaluation, so to help quickly identify them they have been marked with a '[K]' in Figures 5 to 6 and 7 to 10. However, these are subjectively classified and large-scale synoptic forcing in other nodes, for instance due to offshore cyclones, does not preclude the continued occurrence of katabatic winds, hence SOM states other than those marked with a '[K]' are still likely to contain a katabatic influence.





### 3.2.2 Casey

With the lowest Pearson correlation coefficient ($r$) across reanalyses, surface winds observed at Casey station are poorly represented (Figure 7), especially in JRA-55 and MERRA-2 in which the topography of the Law Dome is highly smoothed (Figure 2). Large differences between the observed and reanalysed wind direction are common at Casey (Figures 7). One standout is node (1, 2), a state in which the pressure contours are aligned to favour some flow around the Law Dome. $r$ values for node (1, 2) are higher than for other nodes (Figure G1a) largely due to better representation of wind speed variability above 310 15 ms$^{-1}$. In JRA-55, for instance, there is considerable overestimation of surface wind speeds below 10 ms$^{-1}$ and large spread where the orographic flow blocking is unrepresented, shifting abruptly to consistent underestimation but improved correlation above 10-15 ms$^{-1}$. This large negative bias at elevated wind speeds is present in the other two reanalyses and is in fact greatest in ERA5 (Figure 7f).

Reanalysis wind profiles at Casey (Figure 7) exhibit an excessively strong and deep easterly (negative u) jet at 300-400 m 315 height above ground level (a.g.l.) which is absent in the observations and particularly pronounced when a large-scale pressure gradient is present and oriented to allow flow around the Law Dome (node (1, 2), Figure 7f, l, r). This bias is largest in MERRA-2, consistent with surface wind observations (Figure 7l). Wind speed correlation coefficients improve with height across most nodes. Temperature profiles in the lowest 1000 m are most realistic in ERA5 and are generally biased low in MERRA-2 and JRA-55, especially for the strong synoptic forcing node (1, 2). The temperature profile observed at Casey is often characterised 320 by a near-surface inversion layer in the lowest 100-200 m (see Figure F1) topped by a constant lapse rate which may not be well captured in the coarser products.





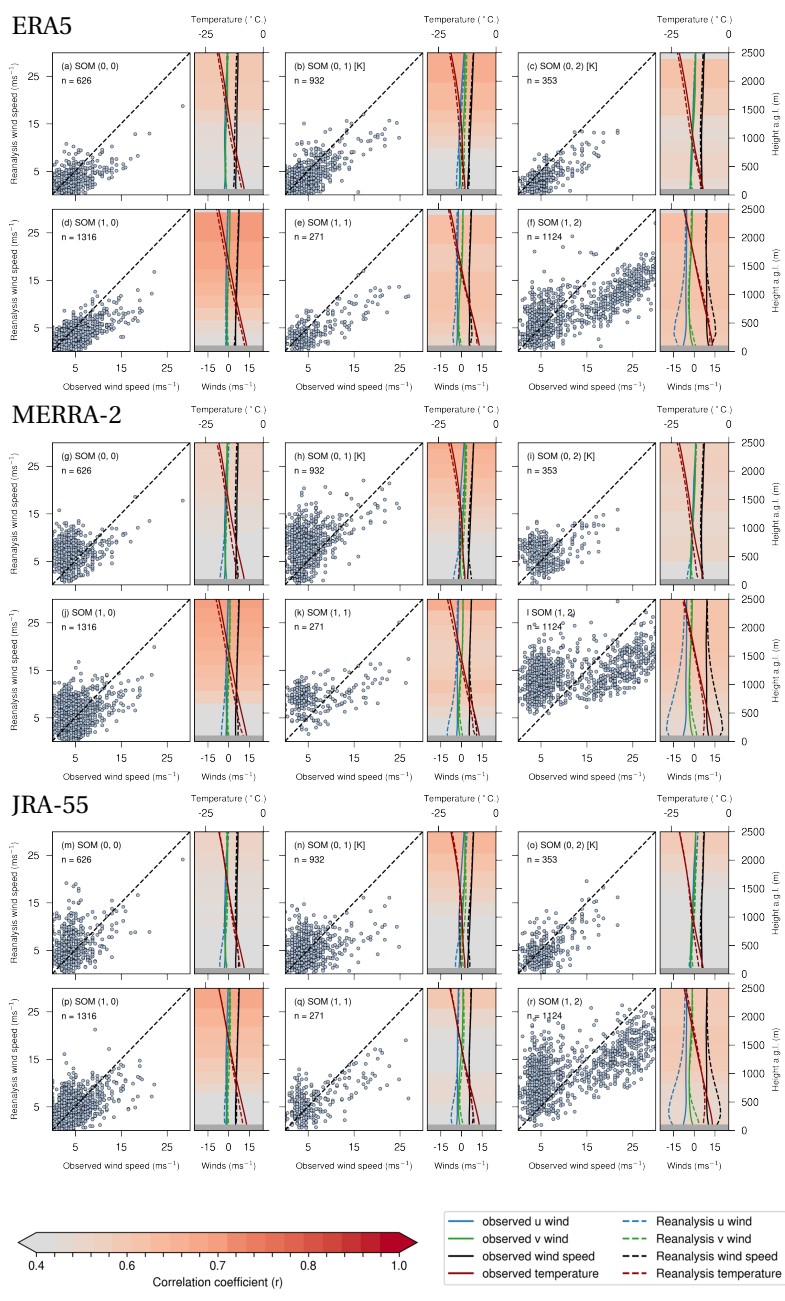

**Figure 7.** Reanalysis-observation wind intercomparison by SOM node at Casey for (a-f) ERA5, (g-l) MERRA-2 and (m-r) JRA-55 for the period 2010-2017. For each node is plotted: (left) scatterplots of observed vs reanalysis 10 m wind speeds and (right) vertical profiles of mean wind speed (black), u wind (blue), v wind (green) and temperature (red) from observations (solid) and reanalysis (dashed). Shading in the background of each profile indicates the correlation coefficient between observed and reanalysis wind speed at that level. Nodes with limited synoptic forcing but favourable conditions for katabatic forcing are labeled with a [K].





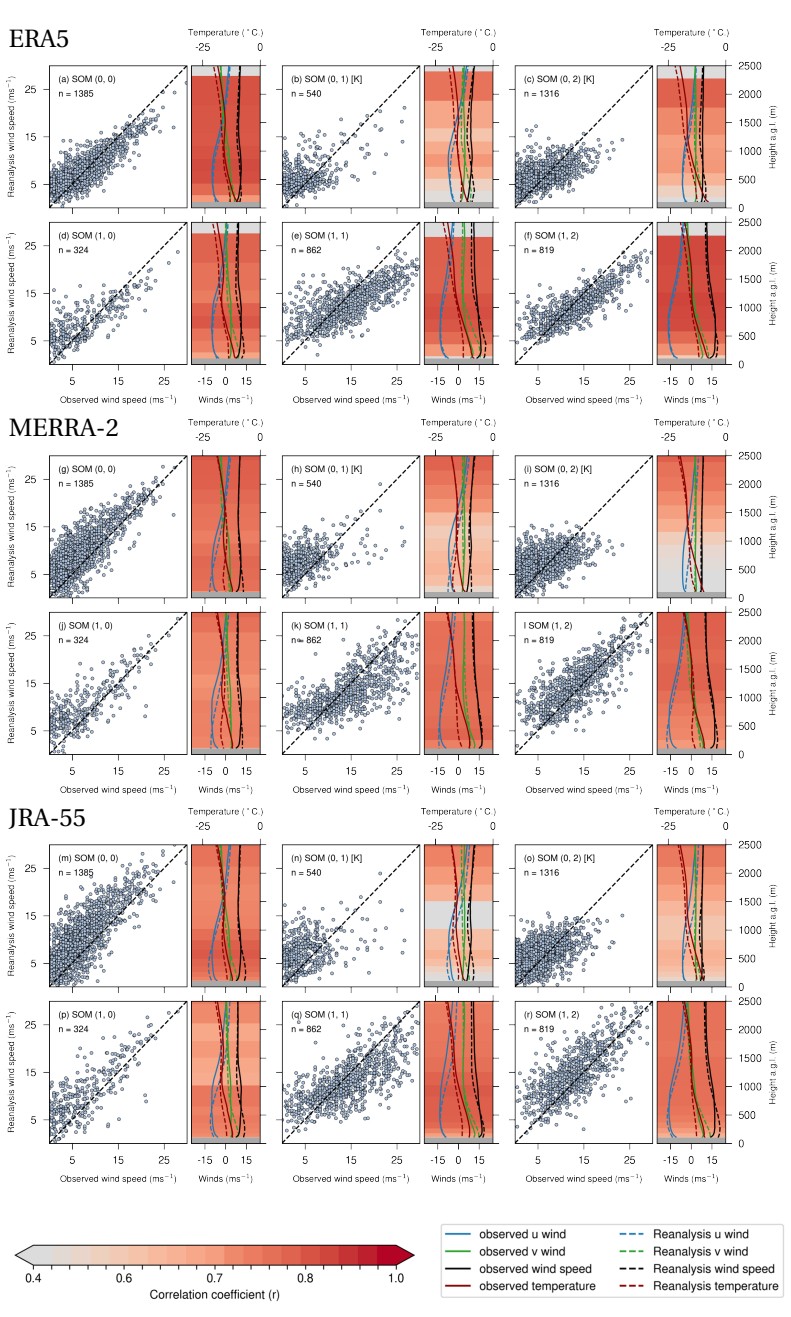

**Figure 8.** As in Figure 7 but for DDU.



### 3.2.3 Dumont d'Urville

At Dumont d'Urville (DDU), 10 m reanalysis winds are more closely aligned with observations than at Casey but performance characteristics vary considerably by SOM state (Figure 8 scatterplots). ERA5 has a systematic negative bias at high wind speeds
(Figures 8a, e and f). By contrast, MERRA-2 and JRA-55 exhibit quite distinct performance regimes at high wind speeds (for example compare Figures 8g and k as well as Figures 8m and q), consistently overestimating wind speeds when the pressure contours favour a more northerly flow (nodes (0, 0) and (1, 2) in Figures 8g, m and 8l, r respectively) but underestimating it when the observed low-level flow has a stronger southerly component (node (1, 1) in Figures 8k and q). Root mean square error (RMSE) is also slightly higher for all reanalyses in node (1, 1) compared to node (0, 0) and (1, 2) (Figure G1b).
States favouring katabatic forcing ((0, 1) and (0, 2), especially) are characterised by a reduced interquartile range (IQR) with respect to the observations in all three reanalyses (Figure G1b), resulting in overestimated low wind speeds and underestimated high wind speeds. Correlation coefficients ($r$) are also reduced for these states, in part due to the smaller range of wind speeds. As with Casey, $r$ values are consistently highest in ERA5 across nodes.

Observed DDU wind profiles exhibit a deep easterly jet accompanied by a shallower southerly layer, both of which are
strongest in nodes (1, 1) and (1, 2) when the synoptic pressure gradient is highest (Figure 8). Correlation coefficients in the lowest 2500 m are higher at DDU compared to Casey, consistent with surface wind data. The core of the easterly winds in jet regimes is similar to observations in all three reanalyses (Figures 8 nodes (1, 1) and (1, 2)), with the main difference between reanalyses being in the height of the jet. The observed jet during the strongest synoptic forcing in node (1, 2) is a deep but uniform feature between 200 and 1000 m a.g.l. whereas the reanalyses represent it with greater shear above and
below a jet core. The jet structure appears to be more realistic in ERA5 for katabatic (0, 1) and low-wind synoptic (0, 2) states compared to MERRA-2 and JRA-55 which position the core of the easterlies several hundred metres too close to the surface and subsequently have considerably lower $r$ values in the lowest 1000 m compared to other nodes. The structure of the meridional winds at DDU is a point of disagreement between reanalyses; in ERA5 the southerly layer is deep and excessively strong in jet cases (Figure 8e, f) whereas in JRA-55 the structure is realistic but the southerlies are too consistently strong
(Figure 8m-r).

Compared to Casey, a more complex temperature profile is observed at DDU. Although not clear from the averaged temperature profiles, near-surface inversions are again common (Figure F1) but they are also accompanied by inversions at altitudes above 1000 m a.g.l., generating a distinct reverse-S shape in the temperature profile which is present in reanalyses but is smoothed and varies considerably in vertical structure between dataset. Substantial negative temperature biases occur in the
lowest 1000 m in JRA-55 and MERRA-2, reaching 5 K at 100 m a.g.l. (Figures 8g-l and 8m-r).




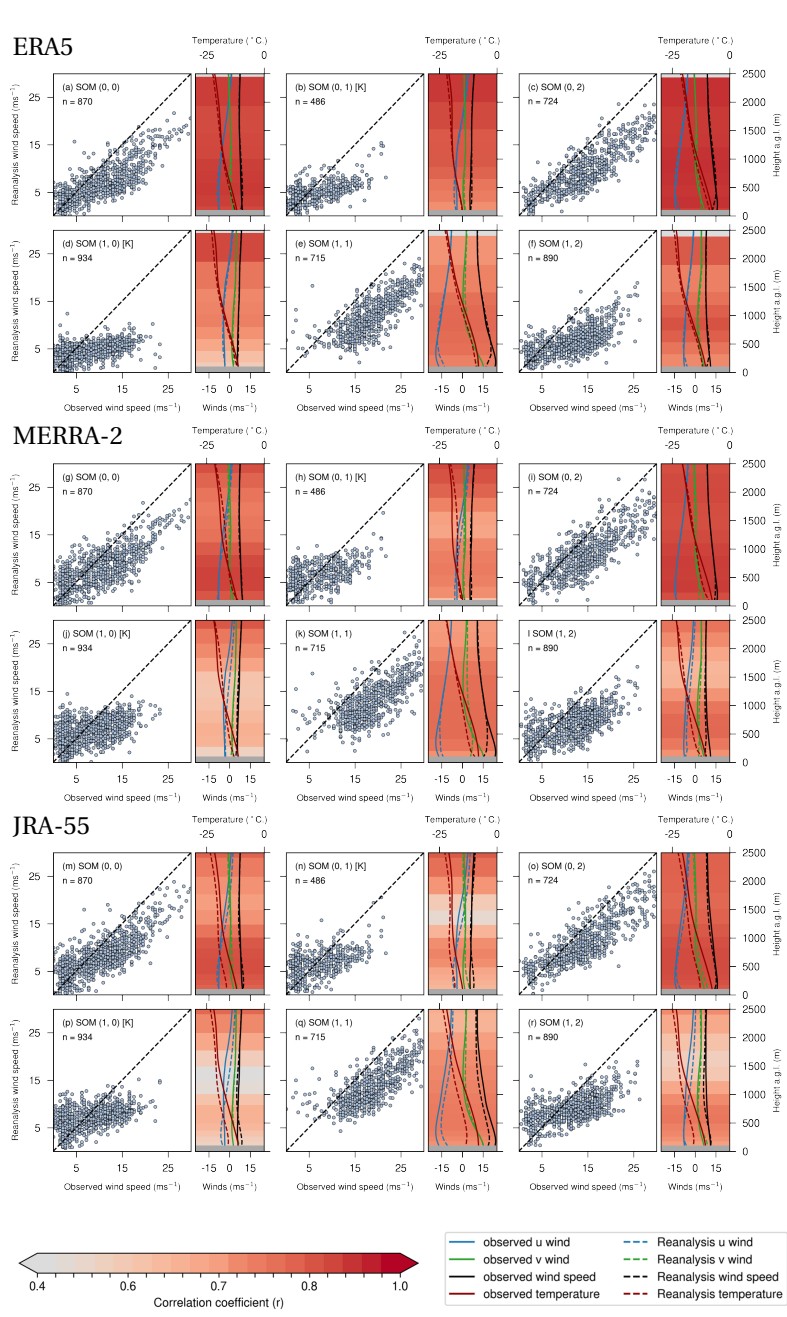

**Figure 9.** As in Figure 7 but for Mawson.



### 3.2.4 Mawson

Compared to other stations, performance statistics differ little between reanalyses at Mawson (Figure G1c), though as before ERA5 has marginally higher $r$ in all nodes. All three underestimate the highest wind speeds, for example in node (1, 1) (Figures 9e, k, q). This underestimation of elevated wind speeds at Mawson is also a feature of the Met Office Unified Model (UM)
which was not rectified by switching to a very high horizontal or vertical grid resolution (Orr et al., 2014).

As at DDU, katabatic states at Mawson ((0, 1) and (1, 0)) are characterised by a considerably reduced IQR of reanalysis wind speeds with respect to observations (Figure G1c); when an intense pressure gradient is absent, reanalyses appear to fall into a regime of underestimating the large local wind speed variability. One interpretation is that there are two distinct flow regimes at Mawson. If the large-scale pressure gradient is strong enough, the onshore and offshore wind field become continuous,
whereas when it is weaker terrestrial (likely katabatic) winds are cut off abruptly at the coast and the representation of the winds becomes sensitive to the placement and turbulent characteristics of the katabatic jump.

The easterly jet at Mawson is similar to DDU (Figure 9), peaking with high synoptic pressure gradients in states (0, 2) and (1, 1) and accompanied by a shallow southerly layer. As with DDU, correlation coefficients against vertical wind measurements vary by SOM node at Mawson. Interestingly, high and consistent $r$ values are found throughout the 2500 m profile for each
of the three reanalyses in node (0, 2) during which the pressure gradient is high but the southerly layer is shallow as a result of the orientation of the low pressure centre offshore, directing geostrophic flow of maritime origin towards Mawson station. By contrast, when a deeper and stronger southerly layer prevails in node (1, 1), the $r$ values are reduced. This may relate to the occurrence of temperature inversions which are poorly resolved by the reanalyses (Figure F1). All three reanalyses underestimate the strength of the u-wind in strong jet cases (1, 1), although the structure of the v-wind and height of the core
of the low-level jet is more realistically represented in JRA-55.

The temperature profile at Mawson is comparable to DDU in its reverse S shape. Again, this is a feature that varies greatly between reanalyses; the structure is most realistic in JRA-55 and ERA5. This may explain why the jet structure (with an elevated easterly core and lower southerly core) is also more realistic in those reanalyses compared to MERRA-2 for which this structure in the vertical wind profile is not seen. As with other stations, JRA-55 and MERRA-2 exhibit a cold bias in the
lowest 1000 m, with that of JRA-55 equalling or exceeding 5 K at the surface.





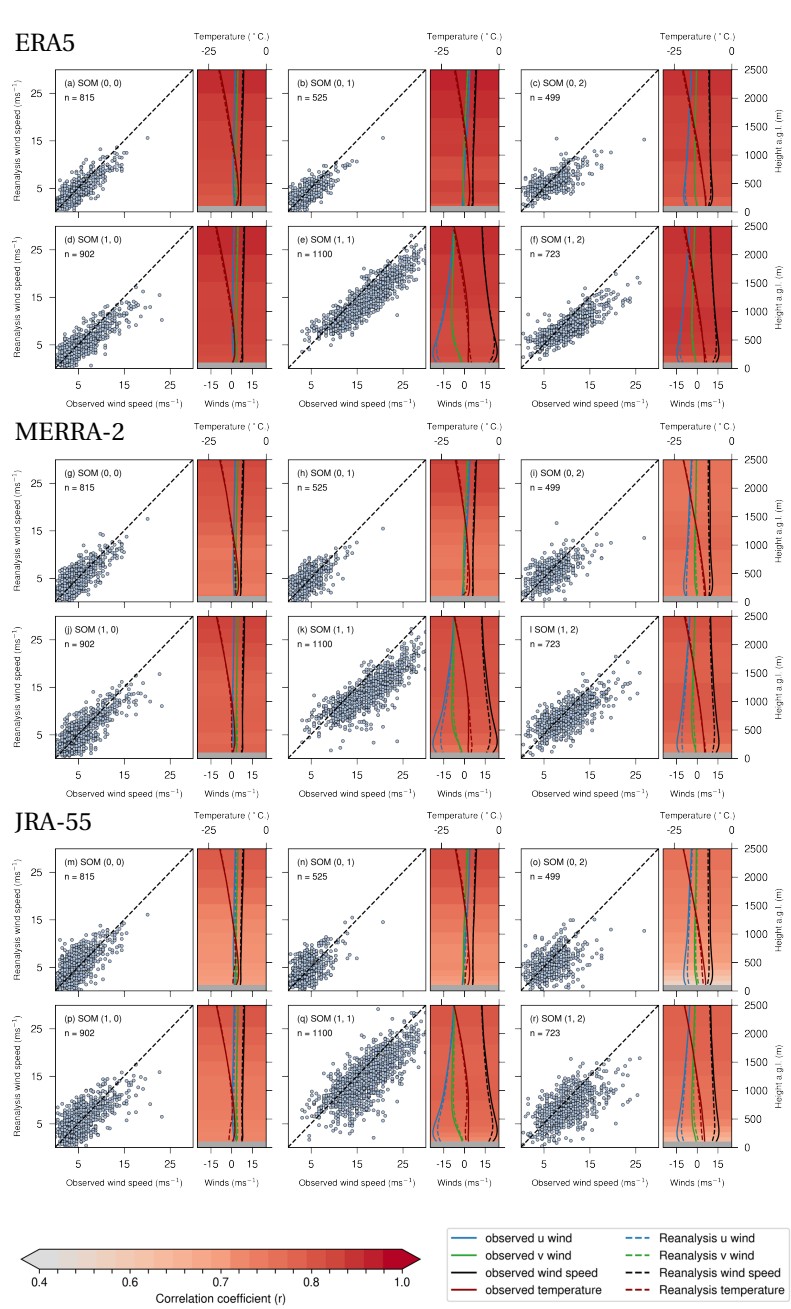

**Figure 10.** As in Figure 7 but for Neumayer.





### 3.2.5 Neumayer

As described in Section 2.3.1, Neumayer has quite different local terrain characteristics compared to other stations, being situated on a flat ice shelf with comparably small orographic height differences between reanalyses and reality. Near-surface wind disagreements are much smaller at Neumayer than at the other stations considered here (Figure 10). As a result, the lowest

mean RMSE values are found here (Figure G1d). ERA5 appears to be the best performing reanalysis by a larger margin than at Mawson. By contrast, lower $r$ values and higher RMSE are found for all SOM nodes in JRA-55 (Figure G1d). There is little discernible difference in the scatterplot characteristics between SOM nodes at Neumayer.

Consistent with better reanalysis performance against surface station winds, both reanalysis wind profiles and reanalysis temperature profiles (Figure 10) compare relatively favourably with Neumayer upper air data. ERA5 and MERRA-2 stand out

here with consistently high wind speed $r$ values across SOM nodes while JRA-55 exhibits some degradation in fidelity in the lowest 500 m, especially for nodes (0, 2) and (1, 2) (Figures 10o and r), both of which are subject to low-level flow passing over the complex coastal terrain east of Neumayer. The jet structure is well represented, including a shallow southerly flow. However, the core of the easterly jet, seen most clearly in nodes (1, 1) and (1, 2), is consistently underestimated by all three datasets. Baroclinicity and the coastal slopes nearby are thought to introduce ageostrophic motions at Neumayer (Kottmeier,

1986; Klöwer et al., 2014) though the katabatic pressure gradient force is not as large over the ice shelf as further inland (van den Broeke and van Lipzig, 2003). Gorodetskaya et al. (2020) show that a slight negative ERA5 wind speed bias is also a characteristic of low-level jets occurring in atmospheric river conditions at Neumayer; our analysis indicates that this bias applies more generally to strong wind conditions, for example as in node (1, 1).

## 4 Discussion

In this paper, we have evaluated the representation of Antarctic coastal winds in three current reanalyses. The analysis aims to quantify representation of the mean state and short-term variability of coastal winds, making use of the few observational datasets available within the coastal easterly domain. There is a great deal in common between reanalyses in the representation of time-mean coastal winds and their variability. These similarities and the differences which do exist point to some critical processes which are important for representation of the coastal easterlies.

Firstly, the representation of orography is especially important at the coast and has a large impact on reanalysis performance. As shown in Section 2.3.1, the slope of the coastal orography in reanalyses is smoothed, leading to large differences in surface height at the latitudes of most coastal stations, with the exception of Neumayer where the local terrain is flat. At Casey, the reanalysis performance is poor across the board, especially for the coarser MERRA-2 and JRA-55, suggesting the local orography (in particular the Law Dome) has a large effect on the winds, including those well above the surface layer. Whereas

reanalysis winds are quite realistic at Neumayer, correlation coefficients only improve at Casey when the large-scale flow is relatively unimpeded by the Law Dome. The comparison with ASCAT also points to a critical role for orography in the coastal margins; reanalysis performance with respect to ASCAT declines with increasing proximity to the coast and biases are especially large close to complex orography such as the Transantarctic Mountains, Law Dome and around Enderby Land.





Orographic processes such as barrier and tip jet formation are likely to be important in the Antarctic coastal margins and
sources of marine wind speed biases in reanalyses.

A second and related theme is the difficulty in characterising low-wind states dominated by katabatic forcing when a strong
large-scale synoptic pressure gradient is absent (nodes marked with a '[K]' in Figures 7, 8 and 9). This is clearest at DDU and
Mawson which are subject to semi-permanent katabatic winds. Vignon et al. (2019) show how a typical feature of ERA-Interim
and ERA5 is an underestimated range between the 5th and 95th percentiles of wind speed in the lowest 200-300 m. Our analysis
indicates this effect may be greatly enhanced during these [K] states; at DDU this mostly appears as an overestimate of low
wind speeds whereas at Mawson high wind speeds are underestimated. The exact source of this misrepresentation is not clear
from this analysis but the implication is that winds in the coastal margins are more variable than reanalyses suggest. A caveat
of this analysis is that katabatic winds are a highly terrain-dependent and locally variable phenomenon at the coast; the sign of
the bias is quite sensitive to the collocation. Furthermore, the contribution of katabatic flow to maritime winds responsible for
coastal currents remains unquantified, but our analysis indicates that katabatic regimes may be a substantial source of model
error.

A third point of interest is that temperature profiles at coastal stations differ between reanalyses, and although this research
focuses on winds these differences could be important for the coastal easterlies. A realistic zonal jet feature does occur in the
lowest 1500 m in reanalyses during states with strong synoptic forcing (except at Casey where it is greatly overestimated), but
MERRA-2 and JRA-55 exhibit substantial cold biases in approximately the lowest 1000-1500 m. A MERRA-2 cold bias in
both East and West Antarctica was also found when compared with observations from the 2010 Concordiasi field campaign
(Ganeshan and Yang, 2019) and against coastal surface stations (Jones et al., 2016; Gossart et al., 2019). Reduced wind
speed correlation coefficients at height, especially in MERRA-2 and JRA-55 during low-wind katabatic states, may be in part
related to these temperature biases and in part to the representation of temperature inversions. A dual inversion structure as
shown for Mawson and DDU in Figure F1 is common for the Antarctic coastline (Truong et al., 2020) and poses a challenge
for faithful representation of near-coastal thermodynamic profiles but its role in the representation of wind profiles warrants
further investigation.

## 4.1 Implications for use of reanalysis datasets

Reanalyses are capable of representing important features of the coastal easterlies and their variability, including the existence
of a jet feature peaking during strong synoptic forcing. Many of the differences against observations are consistent between
datasets, including underestimated surface winds above 20 ms$^{-1}$, underestimated interquartile range in states with reduced
pressure gradients where katabatic winds dominate, poor representation at Casey where a local orographic feature plays a
major role and improved performance at Neumayer over relatively flat surface conditions.

There are several reasons why ERA5 may be most useful as a reference dataset in the coastal easterly sector. Firstly, surface $r$
values against ASCAT and station measurements are on average highest for ERA5, indicating the most faithful representation
of variability in the 10 m winds on short timescales. Secondly, higher resolution in ERA5 supports more realistic coastal
orography which is critical in coastal regions; JRA-55 and MERRA-2 coastal orography is not as steep and theory suggests



that the slope is likely to be important for both katabatic winds and for a deeper baroclinic layer in the coastal zone (Fulton et al., 2017). Lastly, the temperature profile appears to be most realistic in ERA5, with JRA-55 and MERRA-2 exhibiting

substantial cold biases in the lowest 1000 m a.g.l. and a decline in wind speed $r$ values near the top of the easterly jet region in some SOM states. An important caveat in using ERA5 is that it still exhibits the large negative time-mean wind speed biases, shown for both ASCAT and station measurements and especially pronounced at high wind speeds. The contribution of this substantial and time-varying bias to long-term trends and wind stress in the coastal sector warrants further investigation but these results suggest that using ERA5 to analyse coastal extremes requires careful consideration of the effect that a systematic

negative bias could have upon the results.

ASCAT observations and sonde data were assimilated into all three reanalyses, meaning only the 10 m wind observations are quasi-independent in our evaluation. The assimilated observations were likely important for reanalysis performance and so further evaluation with unassimilated observations would be valuable. For example, the effect of assimilating new Antarctic dropsonde observations from the Concordiasi field experiment on ECMWF forecasts was found to be larger away from the

coastal regions where most of the existing radiosondes are launched (Boullot et al., 2016) and additional sonde observations from the Year of Polar Prediction improved wind speed forecasts over West Antarctica (Bromwich et al., 2022).

## 5   Conclusions

The representation of Antarctic coastal easterlies in the ERA5, MERRA-2 and JRA-55 reanalyses has been evaluated by comparison with in-situ and satellite observations, including summertime ASCAT observations over the marine sector and

observations from four coastal stations. For each station, the sensitivity of reanalysis performance to changes in the driving flow regime was analysed using SOMs derived from ERA5 MSLP fields.

A comparison with ASCAT scatterometer winds shows that wind speed variability within the region of the coastal easterlies is well represented by the reanalyses, with correlation coefficients of 0.91 in ERA5, 0.89 in MERRA-2 and 0.85 in JRA-55. Generally, surface wind speeds offshore are biased slightly low across the whole coastal easterly sector in ERA5 and MERRA-

2 whereas JRA-55 biases are closer to zero on average but have large local variations. Correlation coefficients decrease with proximity to the coast and the ASCAT fields reveal near-coastal wind features (especially close to complex orography) whose magnitude is not well captured by the reanalyses. All three reanalyses exhibit larger negative biases relative to scatterometer data at wind speeds exceeding 20 ms$^{-1}$.

Reanalysis performance against observations is much more sensitive to different flow regimes (characterised with SOMs)

at coastal slope stations (in particular Mawson and DDU) than over the flatter terrain of Neumayer. Time series correlation coefficients ($r$) between both surface and upper air wind speeds and collocated reanalysis wind speeds are consistently higher in states with strong synoptic forcing and especially when low-level flow is directed onshore. By contrast, $r$ values consistently drop for states with limited synoptic forcing but favourable conditions for katabatic forcing. This is in part due to the lower wind speeds but also because the reanalyses' interquartile range is lower than observations at both Mawson and DDU during

those states. Performance against observations from Casey is poor due to the effect of the Law Dome on the low-level flow.





Representation of orography is likely critical to variations in reanalysis fidelity by state and station. MERRA-2 and JRA-55 have large cold biases in the lowest 1500 m which may be important for the realism of low-level coastal easterlies.

Overall, ERA5 has the highest surface wind $r$ values compared to both station measurements and ASCAT over the coastal easterly sector. High $r$ values are generally mirrored by low RMSE. A more realistic temperature profile, better characterisation
of variability and more faithful representation of orography gives greater confidence in ERA5 as a benchmark dataset for the coastal easterlies. It should be emphasised, however, that performance is similar between reanalyses and many key deficiencies are shared, including relatively large biases against ASCAT near the coast and at high wind speeds, poor performance close to the Law Dome and underestimation of the wind speed variability when synoptic forcing is reduced and katabatic forcing is important.

These results shed light on some of the challenges associated with evaluating model-based datasets in the Antarctic coastal region and underscore the role of orography. Furthermore, they demonstrate the usefulness of an evaluation approach designed to interrogate reanalysis or model performance under varied atmospheric states, given the high variability of the coastal sector. This SOM-based methodology offers an approach to reanalysis evaluation that is process-oriented and could be applied to other reconstruction datasets or assessment of climate models. In future work the insights from this analysis will be used to
guide regional model sensitivity experiments and to help constrain future projections of the coastal easterlies in coupled climate models.

*Data availability.* ERA5 reanalysis data can be accessed from the Climate Data Store at https://cds.climate.copernicus.eu/cdsapp#!/home (doi: https://doi.org/10.24381/cds.adbb2d47 and https://doi.org/10.24381/cds.bd0915c6). MERRA-2 can be accessed from GES DISC at https://disc.gsfc.nasa.gov/datasets?project=MERRA-2. JRA-55 can be accessed from the Research Data Archive at https://rda.ucar.edu/datasets/ds628.0/.
C-2015 ASCAT data are produced by Remote Sensing Systems and sponsored by the NASA Ocean Vector Winds Science Team. Data are available at http://www.remss.com. Thanks to ICDC, CEN, University of Hamburg for data support (https://www.cen.uni-hamburg.de/en/icdc/data/atmosphere/windvector-ascat.html). Radiosonde data from Neumayer station is available online at https://doi.pangaea.de/10.1594/PANGAEA.874564. Other radiosonde datasets should be requested directly from the institutes referenced in the Acknowledgements. SCAR READER data are available online at https://www.bas.ac.uk/project/reader/.

## Appendix A: Collocation sensitivity test

For the comparison with station measurements, all results were recalculated after shifting the collocated reanalysis gridpoint one point to the north and south.

Although this shift causes a slight degradation in correlation coefficients for all reanalyses, stations and SOM nodes, the main results reported in Sections 3.2.2 to 3.2.5 are robust, except for a large impact upon the value of the bias for wind speeds
below 10 ms$^{-1}$ in states and at stations where the synoptic forcing is reduced but katabatic forcing is favoured. Low-wind states in steep-slope coastal regions therefore appear to be highly sensitive to collocation. For example, in ERA5 for SOM node (0, 2) at DDU the bias is 0.77 ms$^{-1}$, but shifting the collocated point south increases this to a 5.19 ms$^{-1}$ bias (with a





similar effect on low winds in other states) and, conversely, shifting it north gives a -0.9 ms$^{-1}$ bias. Large wind speed gradients occur at the Antarctic coast, especially during katabatic-dominated states, meaning winds upslope (south) of surface stations
may be much stronger than those just offshore (north). Estimates of reanalysis or model bias in the representation of katabatic winds based upon point collocation with coastal station measurements should therefore be treated with caution. Our focus is primarily upon differences between nodes, stations and reanalyses.

**Appendix B:  Selection of SOM configuration**

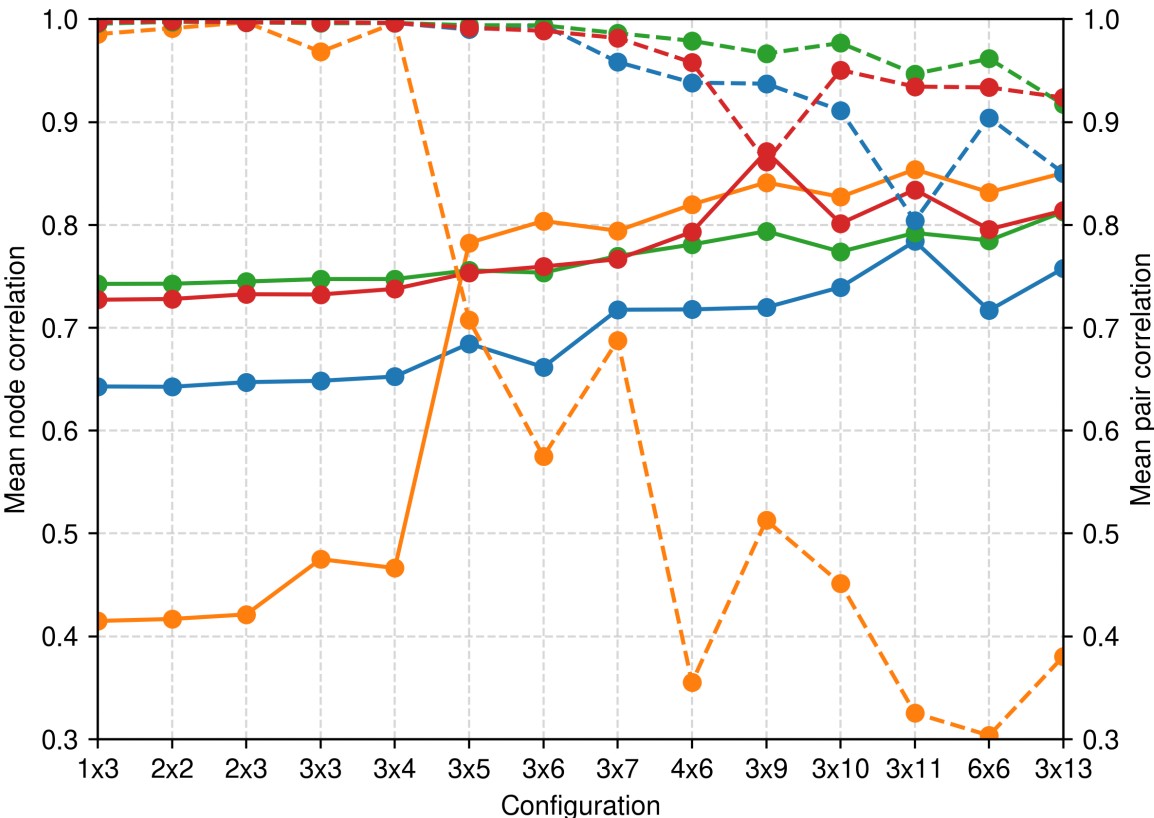

**Figure B1.** Indicators of SOM realism and node uniqueness for different configurations (grid sizes). Solid lines are the mean node correlation and dashed lines the mean pair correlation for Casey (green), DDU (red), Mawson (blue) and Neumayer (orange).




A variety of SOM configurations were tested (Figure B1). Node correlation and pair correlation were calculated based on the approach of Gibson et al. (2017). Node correlation is defined as the mean correlation between the data points comprising individual SOM nodes and the data points of timesteps to which they were assigned (i.e. a reanalysis timestep is assigned to a SOM node based on the mean squared difference and then the correlation is calculated between all data points in that timestep and all data points in the SOM node). The pair correlation is defined as the mean correlation between all possible pairs of SOM nodes. The node correlation is therefore intended as a measure of how realistic SOM nodes are (larger values are desirable) whereas pair correlation is a measure of how similar they are to each other (lower values are desirable). It should be noted that pair correlations near one for configurations with few nodes are a result of the background large-scale conditions of the Antarctic; high pressure over the continent and low pressure offshore within the circumpolar trough, with this dipole not differing greatly between nodes in Figure 5 despite varying patterns. Only at Neumayer is there some evidence in Figure 5 of this dipole reversing.

SOMs for Casey, Mawson and DDU were found to be relatively insensitive to the number of nodes, with very little change in the above statistics until around a $3 \times 6$ configuration (Figure B1). Neumayer, on the other hand, is much more sensitive to the number of SOM nodes. This is likely because it is situated within a region of high MSLP variability (near the northern edge of the peak 12-hourly wind speed standard deviation in Figure 1) compared to other stations where the local flow is much more constrained by orography. It could be argued then that a $3 \times 5$ configuration would be preferable for Neumayer. However, the performance of the reanalyses was found not to vary by node at Neumayer even for SOM grids with many nodes. For ease of interpretation, therefore, we opt for a grid with fewer nodes. Inclusion of additional SOM nodes at other stations was not found to add much value to the analysis except in distinguishing different varieties of extreme wind events, which is beyond the scope of the current work but may be useful for a more targeted analysis in future.

Overall, A SOM configuration of $2 \times 3$ was deemed to be capable of capturing critical differences in reanalysis performance (associated with katabatic and synoptic states as well as variations in flow orientation) without being too large for the reader to interpret easily.





**Appendix C: Calculation of katabatic index**

The katabatic index used in this paper is calculated as:

$$K = \frac{g}{\theta_0} \Delta_\theta \sin \alpha \tag{C1}$$

where $g$ is acceleration due to gravity and $\alpha$ is the angle of the slope in the direction for which $K$ is evaluated. The terms $\theta_0$
and $\Delta_\theta$ are, respectively, the 'background' potential temperature if long-wave cooling of the surface layer were neglected and
the departure from this background. These terms therefore represent the perturbation to the lapse rate near the surface assumed
to drive katabatic flow. Here, the value of $\theta_0$ is approximated by assuming a linear background lapse rate between 600 hPa and
the surface and extrapolating based on the potential temperature gradient between 500 and 600 hPa down to the ERA5 surface

height. $\Delta_\theta$ is then calculated by subtracting the observed surface potential temperature (linearly extrapolated from the pressure
level closest to the surface to the ERA5 surface height) from $\theta_0$.





## Appendix D: Seasonal breakdown of ASCAT results

**Figure D1.** Mean ERA5-collocated ASCAT 10 m wind speed (shaded) and wind field (wind vectors) for the period 2010-2017 for individual calendar months from (a) January to (l) December. The same hatching of missing data, orography contours and demarcation of the coastal easterly zone is used as in Figure 4.







**Figure D2.** Mean ERA5 minus ASCAT 10 m wind speed for the period 2010-2017 for individual calendar months from (a) January to (l) December. The same hatching of missing data, orography contours and demarcation of the coastal easterly zone is used as in Figure 4.



**Figure D3.** As in Figure D2 but for MERRA-2.





**Figure D4.** As in Figure D2 but for JRA-55.





## Appendix E: ASCAT zonal and meridional components

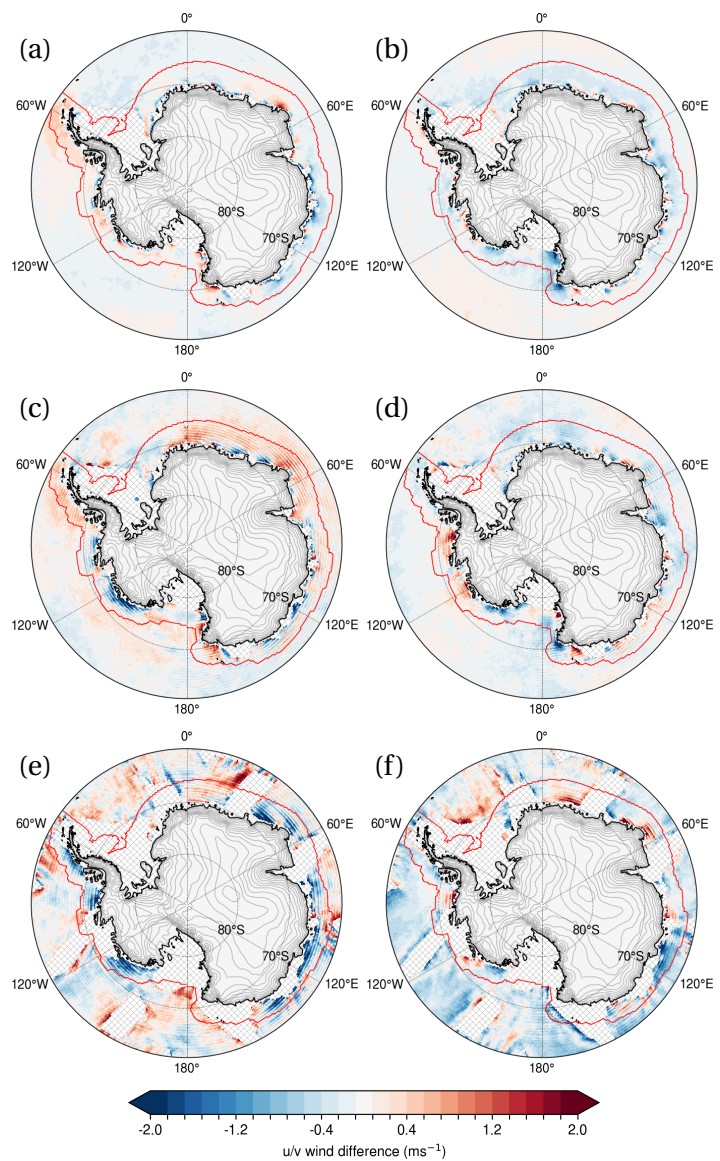

**Figure E1.** Mean reanalysis minus ASCAT 10 m zonal wind (left column) and meridional wind (right column) for the period 2010-2017 for (a-b) ERA5, (c-d) MERRA-2 and (e-f) JRA-55. Hourly data are used for ERA5 and MERRA-2 with six-hourly from JRA-55. Data points which are rain flagged or have a GMF-matchup flag value over 2 are not included and pixels with fewer than 50 ASCAT-reanalysis collocations are masked. Orography at 300 m intervals (from the ERA5 invariant fields) is marked with grey contours. The region within the red contour is the coastal easterlies domain, identified where either the ERA5 2010-2017 mean zonal wind is less than zero or the location is within a 12-gridbox buffer drawn from the coast.





## Appendix F: Temperature inversions

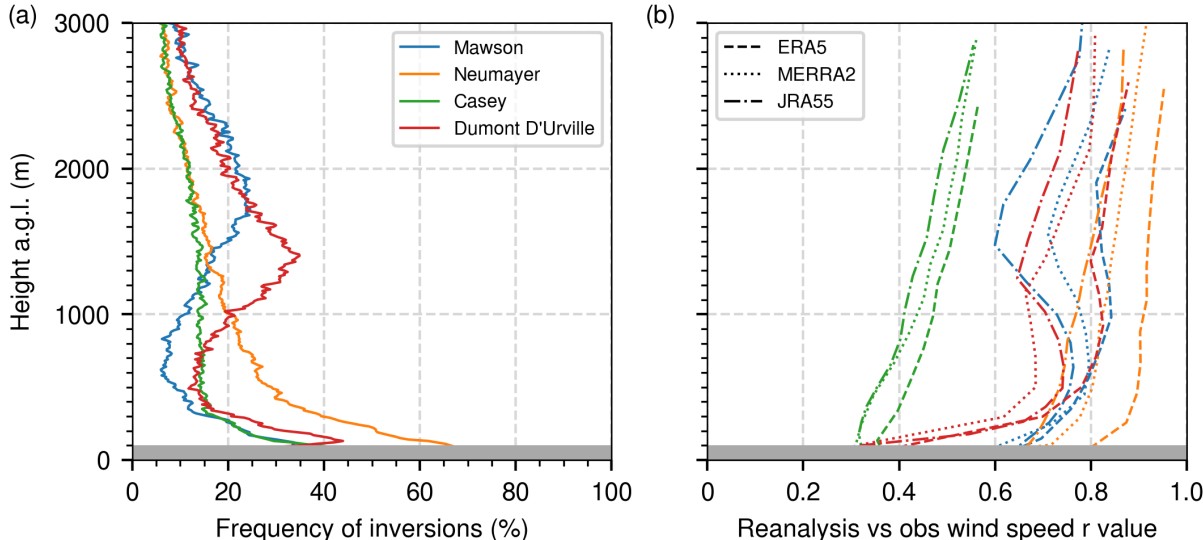

**Figure F1.** (a) Frequency of temperature inversions with height at Casey (green), DDU (red), Mawson (blue) and Neumayer (orange), with temperature inversions defined as occurring where temperature increases with height for a given 10 m level in the sonde observations for the period 2010-2017. (b) Mean correlation coefficient between observed and reanalysis wind speeds with height for the period 2010-2017, using the same colour scheme for ERA5 (dashed), MERRA-2 (dotted) and JRA-55 (dash-dotted).





**Appendix G: Detailed SOM state statistics**

| | | ERA5 | | | MERRA-2 | | | JRA-55 | | |
|---|---|---|---|---|---|---|---|---|---|---|
| **(a)** | r | 0.539 | 0.652 | 0.746 | 0.228 | 0.378 | 0.299 | 0.355 | 0.324 | 0.623 |
| | RMSE | 2.882 | 2.811 | 3.413 | 4.278 | 5.400 | 4.143 | 3.864 | 3.565 | 3.172 |
| | MB | -1.166 | -0.972 | -2.269 | 2.228 | 3.701 | 1.121 | 1.115 | 0.082 | -0.691 |
| | IQRB | 0.123 | 0.101 | -0.726 | 1.429 | 1.360 | 0.204 | 1.744 | 0.196 | -0.423 |
| | r | 0.642 | 0.772 | 0.849 | 0.286 | 0.413 | 0.662 | 0.417 | 0.630 | 0.731 |
| | RMSE | 2.733 | 3.986 | 7.905 | 3.703 | 5.586 | 8.629 | 3.099 | 4.087 | 7.687 |
| | MB | -1.370 | -1.948 | -4.186 | 0.909 | 2.810 | 1.132 | 0.048 | -0.259 | -0.349 |
| | IQRB | -0.670 | -1.202 | -10.068 | 1.343 | -1.508 | -12.756 | 0.205 | 0.106 | -10.423 |
| **(b)** | r | 0.843 | 0.566 | 0.516 | 0.803 | 0.460 | 0.450 | 0.774 | 0.348 | 0.470 |
| | RMSE | 2.863 | 4.002 | 3.484 | 4.228 | 4.912 | 3.896 | 4.857 | 5.061 | 3.715 |
| | MB | 0.707 | 1.638 | 0.769 | 2.838 | 2.898 | 1.424 | 3.304 | 2.428 | 0.867 |
| | IQRB | -1.893 | -2.533 | -2.415 | 0.129 | -1.537 | -1.815 | 1.069 | -0.866 | -1.489 |
| | r | 0.785 | 0.749 | 0.833 | 0.782 | 0.718 | 0.784 | 0.782 | 0.721 | 0.763 |
| | RMSE | 3.878 | 5.002 | 3.588 | 4.504 | 5.006 | 3.993 | 4.668 | 4.993 | 4.394 |
| | MB | 1.227 | -1.482 | -0.871 | 2.583 | -1.041 | 1.432 | 2.707 | -1.099 | 1.929 |
| | IQRB | -2.514 | -3.287 | -2.841 | -2.220 | -3.148 | -0.832 | -1.577 | -2.928 | -0.621 |
| **(c)** | r | 0.800 | 0.615 | 0.859 | 0.770 | 0.567 | 0.817 | 0.784 | 0.554 | 0.847 |
| | RMSE | 5.143 | 4.550 | 6.055 | 4.649 | 4.247 | 5.344 | 4.478 | 4.267 | 4.958 |
| | MB | -3.221 | -2.029 | -4.683 | -2.090 | -0.752 | -3.341 | -1.883 | -0.579 | -3.091 |
| | IQRB | -5.973 | -5.676 | -2.172 | -4.685 | -4.223 | -2.114 | -5.628 | -4.752 | -2.047 |
| | r | 0.522 | 0.843 | 0.655 | 0.497 | 0.793 | 0.616 | 0.479 | 0.821 | 0.599 |
| | RMSE | 5.592 | 7.839 | 6.698 | 4.653 | 7.340 | 5.543 | 4.521 | 6.435 | 5.146 |
| | MB | -3.477 | -6.905 | -5.327 | -1.540 | -6.048 | -3.636 | -0.875 | -5.106 | -2.863 |
| | IQRB | -6.271 | -1.088 | -4.131 | -4.576 | -1.790 | -3.356 | -5.939 | -1.601 | -4.209 |
| **(d)** | r | 0.818 | 0.744 | 0.685 | 0.752 | 0.662 | 0.610 | 0.658 | 0.565 | 0.437 |
| | RMSE | 1.630 | 1.715 | 2.484 | 1.895 | 1.991 | 2.669 | 2.346 | 2.365 | 3.371 |
| | MB | -0.229 | -0.296 | -0.597 | -0.002 | 0.099 | -0.408 | 0.335 | 0.219 | -1.029 |
| | IQRB | -0.658 | -0.538 | -2.472 | 0.041 | 0.115 | -1.626 | 0.736 | 0.441 | -0.938 |
| | r | 0.836 | 0.902 | 0.808 | 0.795 | 0.832 | 0.722 | 0.688 | 0.757 | 0.603 |
| | RMSE | 2.072 | 3.407 | 2.936 | 2.185 | 4.290 | 3.127 | 2.673 | 4.182 | 3.767 |
| | MB | -0.659 | -2.491 | -1.504 | -0.172 | -3.103 | -1.223 | -0.181 | -2.224 | -1.577 |
| | IQRB | -1.119 | -1.650 | -1.807 | -0.256 | -2.443 | -1.659 | 0.166 | -0.796 | -1.119 |

**Figure G1.** Performance statistics calculated from reanalysis vs observed 12-hourly 10 m wind speed for (a) Casey, (b) DDU, (c) Mawson and (d) Neumayer. Left three columns correspond to ERA5, middle three MERRA-2 and right three JRA-55, with individual outlined boxes corresponding to SOM nodes as laid out in Figure 5. Statistics shown include (by row) Pearson's correlation coefficient (r), root-mean square error (RMSE, $ms^{-1}$), mean bias (MB, $ms^{-1}$) and reanalysis minus observed interquartile range (IQRB, $ms^{-1}$).

*Author contributions.* TCH led the analysis, interpretation of results and manuscript preparation with contributions from all co-authors. TCH and SB designed and implemented the analysis techniques and developed the analysis code. EV maintains and provided access to the high-resolution radiosonde data.



*Competing interests.* The authors declare that they have no conflict of interest.

*Acknowledgements.* T. Caton Harrison and S. Biri were funded by NERC grant 'Improved projections of winds at the crossroads between Antarctica and the Southern Ocean' (NE/V000691/1 at BAS and NE/V000969/1 at NOC). This work forms part of the Polar Science for Planet Earth program of the British Antarctic Survey. We are grateful to the European Centre for Medium-range Weather Forecasts (ECMWF), the NASA Global Modeling and Assimilation Office (GMAO) and the Japan Meteorological Agency (JMA) for making ERA5, MERRA-2 and JRA-55 available, respectively. For radiosonde data from Casey and Mawson station we thank the Australian Bureau of
Meteorology. For radiosonde data from Neumauyer we gratefully acknowledge König-Langlo (2017) who make the data openly available at the link in the data statement. For radiosonde data from Dumont d'Urville we thank Météo France, the DSO/DOA service for the acquisition and distribution of the data. We are grateful to Steve Colwell for supporting and disseminating the READER dataset. C-2015 ASCAT data are produced by Remote Sensing Systems and sponsored by the NASA Ocean Vector Winds Science Team. Data are available at http://www.remss.com; thanks to ICDC, CEN, University of Hamburg for data support.





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
