# Peer review of "Reanalysis representation of low-level winds in the Antarctic near-coastal region"

_EGUsphere, 2022_

## Referee Comment (RC1)

**Review for Harrison et al. (2022) Reanalysis representation of low-level winds in the Antarctic near-coastal region**

https://doi.org/10.5194/egusphere-2022-693

**General Comments**

The manuscript addresses the quality of near surface winds in three reanalyses at four coastal stations in Antarctica. The study is motivated by the importance of the southern hemisphere polar easterlies to the climate system, especially the coupling between the easterly low level winds that encircle Antarctica, the Antarctic coastal current and Antarctic slope front. The Self Organizing Maps -method is applied to analyze the different weather regimes and their importance to the performance of the reanalyses.

The topic of this study is of importance to the community. It increases the understanding on the sources of low-level wind speed errors in the reanalyses, and gives recommendations on the use of reanalyses. The data is appropriate: the study addresses the newest atmospheric reanalyses, and uses diverse datasets in the assessment (satellite observations and in-situ observations from meteorological stations and radiosondes). A potential data-related weakness is that the ASCAT-observations are assimilated to the reanalyses. However, a large part of the assessment is based on independent in situ observations, so the assimilation of ASCAT winds should not be a crusial problem for the overall results.

Methods are suitable for the study. A wide range of basic statistic is shown. Application of SOM gives good insight in the role of different weather regimes. The SOM method is well described, and the essential question of the number of nodes is discussed in the manuscript and in the supplement.

The overall structure of the text is clear and the quality of the plots is good. However, the manuscript could profit from a more tight focus on the relevant results. This would also allow to narrow down the number of figures, which is currently quite high. I will provide more suggestion on this in the following paragraphs.

The results support well conclusions, and the conclusion are physically sound. With the broad analysis that you have done, you could give even bolder statements in Section 4.1. I will give some suggestions related to this in the following sections.

**Specific Comments**

**SOM methods and the number of nodes**

Could you comment on how the length of the timeseries affected the number of nodes that you selected? Your timeseries is not very long, so I assume that it also had an impact on the number of SOM nodes.

**Number of figures**

There are 10 figures and 1 table in the manuscript, and 8 figures in the supplementary material. Most of the figures contain several panels making them packed with information. In my opinion, the number of figures and subplots could be reduced, without loosing the essential message of the manuscript. I suggest that the authors consider the following changes:

- **Could Table 1 be removed?** It is a good overview of the weather regimes and clearly useful in the analysis phase of the study. However, I don't think that it is necessary in the final manuscript as the same information comes clear through the figures (especially Fig. 5), and the text.
- **Could you select only 2-3 of the SOM nodes to be shown in figures 7-10?** For example, you could select only the three most frequent nodes for each station. This would reduce the subplots from 6 to 3, and make the plots more visible. Also, this selection would allow to keep examples of a katabatic case and a strong wind case, which you discuss the most in the text. You could keep the text almost as it is, as you focus already mostly to these two flow regimes. However, these most frequent nodes don't necessarily capture the strongest cases, so some changes will be required. The most frequent nodes, ie. the nodes to keep would be (according to Fig. 5):
    - Casey, figure 7:       (0,1), (1,0), and (1,2)
    - DDU, figure 8:         (0,0), (0,2), and (1,1)
    - Mawson, figure 9:     (0,0), (1,0), and (1,2)
    - Neumayer, figure 10:  (0,0), (1,0), and (1,1)

**Implications for the use of reanalysis datasets (section 4.1 of the manuscript)**

As you do a thorough analysis of the performance of the reanalysis, you could highlight these results more in section 4.1. For example, you could provide a ranking for the reanalyses similar to what is done in Jonassen et al. 2019, JGR-Atmospheres (https://doi.org/10.1029/2019JD030897). I feel that this kind of assessment could help in drawing overall conclusions on the different performance between the reanalyses. You could do this ranking based on the supplement Figure G1. I provide below an example of the ranking for station Casey. If you like this approach, you could calculate the ranks for each station, and justify your recommendation in section 4.1 with the help of these ranking values.

Example ranking for Casey-station:

*Performance ranks for each node based on Fig G1.*

The reanalysis that performs best for the particular node gets a rank 1, the second best gets 2, and the third 3 (note: Jonassen et al have the ranks the other way round). The ranking is done for node and each metric (each row in this case) separately. The sum of the scores (for each node and reanalysis) is shown in the "total"-row.

| | ERA5 | | | MERRA-2 | | | JRA-55 | | |
|---|---|---|---|---|---|---|---|---|---|
| R | 1 | 1 | 1 | 3 | 2 | 3 | 2 | 3 | 2 |
| RMSE | 1 | 1 | 2 | 3 | 3 | 3 | 2 | 2 | 2 |
| MB | 2 | 2 | 3 | 3 | 3 | 2 | 1 | 1 | 2 |
| IQRB | 1 | 1 | 3 | 2 | 2 | 1 | 3 | 3 | 2 |
| **Total:** | **5** | **5** | **9** | **11** | **10** | **9** | **8** | **9** | **8** |
| R | 1 | 1 | 1 | 2 | 3 | 3 | 3 | 2 | 2 |
| RMSE | 1 | 1 | 2 | 3 | 3 | 3 | 2 | 2 | 1 |
| MB | 3 | 2 | 3 | 2 | 3 | 2 | 1 | 1 | 1 |
| IQRB | 1 | 2 | 1 | 2 | 3 | 3 | 1 | 1 | 1 |
| **Total:** | **6** | **6** | **7** | **9** | **12** | **11** | **7** | **6** | **5** |

*Total ranks for all nodes and metrics.*

The smaller the number the better the overall performance. The best possible score is 24 (reanalysis performing better that the two others in all flow regimes and according to all metrics), and the worst score is 72. You see here also, that for nodes (0,2) and (1,2) JRA performs better than ERA5. JRA-55 is also better than ERA5 in terms of mean bias.

The total scores for Casey are:
- ERA5    38    (as from 5+5+9+6+6+7)
- JRA-55    43
- MERRA-2    62

**Supplementary information**
I don't think you need to have Appendix D at all, because the figures D are mentioned only once on page 6 line 134.

**Technical Corrections:**
- If I'm not mistaken, the time period of the study is mentioned quite late in the text (page 9): could you mention it earlier?
- Caption of Fig 3: I find it hard to understand panels (a) and (b). Could you try to rephrase the caption?
- Page 10, line 226: *"[…] around the coast (Figure 3b)"* -> again, I cannot grasp the idea. Do you mean that ERA5 has the highest correlation anywhere around the coast, no matter what the latitude is?
- Page 11, Figure 4: why does JRA55 have so much more missing data than ERA5 and MERRA-2? Do you mention it somewhere in the text?
- Page 11, line 245-246: *"[…] JRA-55 does not have as consistent a sign of difference in wind speed compared to the other two reanalyses."* -> this is correct, but it seem to me also, that JRA-55 bias pattern is similar to that of ERA5 and MERRA-2. According to Fig 4, it seem that JRA has systematically higher wind speeds than ERA5 and MERRA-2, but the bias pattern is similar. This might be worth mentioning.
- Page 13, Line 284-285: *"[…]nodes (1, 1) and (1, 2) have comparable large-scale pressure gradients but the low-level pressure contours associated with the offshore low are orientated along the coast for (1, 1) whereas in (1, 2) they are oriented more perpendicular to the coast."* -> I think that an essential difference between DDU nodes (1,1) and (1,2) is also that the low pressure center is located on different side of the station.
- Page 14, Table 1: As I mentioned, I don't think that this table is necessary for the final manuscript.
- Page 16, line 300-301: *" they have been marked with a '[K]' in Figures 5 to 6 and 7 to 10. "* -> It is a good idea to mark the katabatic cases in the figures, however, there is a slight risk to confuse K with kelvin-units. Maybe you could use "Kat." ?
- Titles for sections 3.2.2 to 3.2.5 could be more informative. For example "Wind and temperature profiles at XXX" or "Effect of weather regimes on the wind and temperature profiles at XXX"
- Figures 7-10: test the significance of the correlation (eg. student t-test for correlations), mark the significance level on the figure, or mention it in the caption.

---

## Author Response (AR1)

**Reviewer response**

**Response colour code:**

- Authors' response to reviewer
- Text added to revised manuscript
- Text from original unrevised manuscript

All referenced line numbers refer to the revised manuscript with tracked changes.

**Reviewer 1**

**General Comments**

The manuscript addresses the quality of near surface winds in three reanalyses at four coastal stations in Antarctica. The study is motivated by the importance of the southern hemisphere polar easterlies to the climate system, especially the coupling between the easterly low level windsthat encircle Antarctica, the Antarctic coastal current and Antarctic slope front. The Self Organizing Maps -method is applied to analyze the different weather regimes and their importance to the performance of the reanalyses.

The topic of this study is of importance to the community. It increases the understanding on the sources of low-level wind speed errors in the reanalyses, and gives recommendations on the use of reanalyses. The data is appropriate: the study addresses the newest atmospheric reanalyses, and uses diverse datasets in the assessment (satellite observations and in-situ observations from meteorological stations and radiosondes). A potential data-related weakness is that the ASCAT-observations are assimilated to the reanalyses. However, a large part of the assessment is based on independent in situ observations, so the assimilation of ASCAT winds should not be a crusial problem for the overall results.

Methods are suitable for the study. A wide range of basic statistic is shown. Application of SOM gives good insight in the role of different weather regimes. The SOM method is well described, and the essential question of the number of nodes is discussed in the manuscript and in the supplement.

The overall structure of the text is clear and the quality of the plots is good. However, the manuscript could profit from a more tight focus on the relevant results. This would also allow to narrow down the number of figures, which is currently quite high. I will provide more suggestion on this in the following paragraphs.

The results support well conclusions, and the conclusion are physically sound. With the broad analysis that you have done, you could give even bolder statements in Section 4.1. I will give some suggestions related to this in the following sections.

We are grateful to the reviewer for their detailed review, as well as for taking the time to make suggestions with worked examples. We agree with the main critique of needing to tighten the focus of the paper and have addressed these concerns in the responses below.

**Specific Comments**

**SOM methods and the number of nodes**

Could you comment on how the length of the timeseries affected the number of nodes that you selected? Your timeseries is not very long, so I assume that it also had an impact on the number of SOM nodes.

Thank you for raising this: we would like the SOMs for each station to be stable and representative. One test of the effect of the length of the time series we can do is to repeat the analysis of Appendix B for an even shorter time series. Appendix B tests how the realism (how well do the nodes correlate with the actual data points assigned to them?) and uniqueness (how different are the nodes from each other?) varies with different SOM configurations (1x3 up to 3x13). Below we show the results from the original analysis as shown in Figure B1, compared to the results from an analysis with only one year of data (2017):

Casey is shown in green, DDU in red, Mawson in blue and Neumayer in orange. Node correlation is the solid lines (realism, higher is good) and pair correlation is the dashed line (similarity of nodes, lower is good). The results are very similar to what we had for 2010-2017. The node correlations are a bit higher in the one-year case for smaller SOM configurations because the SOMs are quite generalized but individual data points are more likely to fit those generalized SOMs as they're from the same year of data.

We might then also expect that with stable and realistic SOM nodes we would at least be able to get a large enough sample of matched obs/reanalysis data points that they are correlated (significantly), or more specifically that the insignificance of the correlation is not a result of the sample size being too small. For the one year case, the number of nodes really matters for this. Below is a comparison of a very large node array (3x13) for the 8 year period (above) and 1 year period (below) at Mawson. Levels where the obs vs reanalysis correlations of wind speed are significant at the 99% level are hatched: